# Taxonomy and Multigene Phylogeny of *Diaporthales* in Guizhou Province, China

**DOI:** 10.3390/jof8121301

**Published:** 2022-12-15

**Authors:** Si-Yao Wang, Eric H. C. McKenzie, Alan J. L. Phillips, Yan Li, Yong Wang

**Affiliations:** 1Department of Plant Pathology, Agriculture College, Guizhou University, Guiyang 550025, China; 2Key Laboratory of Plant Resources Conservation and Germplasm Innovation in Mountainous Region (Ministry of Education), College of Life Sciences/Institute of Agro-Bioengineering, Guizhou University, Guiyang 550025, China; 3Private Bag 92170, Auckland Mail Centre, Auckland 1142, New Zealand; 4Faculdade de Ciências, Biosystems and Integrative Sciences Institute (BioISI), Universidade de Lisboa, Campo Grande, 1749-016 Lisbon, Portugal

**Keywords:** *Chrysofolia*, *Diaporthe*, *Foliocryphiaceae*, one new genus, *Pseudomastigosporella*, seven new species

## Abstract

In a study of fungi isolated from plant material in Guizhou Province, China, we identified 23 strains of *Diaporthales* belonging to nine species. These are identified from multigene phylogenetic analyses of ITS, LSU, *rpb2*, *tef1*, and *tub2* gene sequence data coupled with morphological studies. The fungi include a new genus (*Pseudomastigosporella*) in *Foliocryphiaceae* isolated from *Acer palmatum* and *Hypericum patulum*, a new species of *Chrysofolia* isolated from *Coriaria nepalensis*, and five new species of *Diaporthe* isolated from *Juglans regia*, *Eucommia ulmoides*, and *Hypericum patulum*. *Gnomoniopsis rosae* and *Coniella quercicola* are newly recorded species for China.

## 1. Introduction

*Diaporthales* is an important and species-rich ascomycetous order in the subclass *Diaporthomycetidae* (*Sordariomycetes*). Despite its cosmopolitan distribution and high diversity with distinctive morphology, this order has received relatively little attention. Currently, the existing classification lists 31 accepted families within the order *Diaporthales* [1], including *Foliocryphiaceae*, *Diaporthaceae*, *Gnomoniaceae*, and *Schizoparmaceae*. Members of *Diaporthales* have a wide range of ecological habitats and numerous modes of nutrition [2]. Excepting the members of *Tirisporellaceae*, most taxa in *Diaporthales* occur in terrestrial habitats. Species in *Diaporthales* form solitary or aggregated, immersed to erumpent, rarely superficial, and orange, brown, or black perithecial ascomata, with short or long necks that are located in stromatic tissues or substrates and with a lack of hamathecium or with few paraphyses [2,3,4,5]. Their asci are unitunicate with a conspicuous refractive ring [5,6]. Their ascospores are diverse in shape, size, and color. The asexual morphs of *Diaporthales* are generally coelomycetous [6], producing acervuli, pycnidial, or synnematal conidiomata and with or without a well-developed stroma. Conidiogenesis is phialidic or rarely annellidic, and conidia are usually unicellular or one-septate [6].

In China, the first monograph for *Diaporthales* referred to *Phomopsis* (=*Diaporthe*), which introduced 133 morphological taxa (including two specialized forma) isolated from 74 familial plants [7]. One diaporthalean pathogen that causes a devastating wilt disease for *Cyathea lepifera* was reported in Taiwan [8]. *Pustulomyces* accommodated in *Diaporthaceae* was revealed by morphology and molecular analyses [9]. Two novel families, *Melansporellaceae* and *Diaporthosporellaceae*, were introduced to accommodate the monotypic genera, *Melanosporella* and *Diaporthosporella*, based on both holomorphic morphology and phylogenetic analysis [10,11], and *Foliocryphiaceae* was established by Jiang et al. [1] to retain *Chrysofolia*, *Foliocryphia*, and *Neocryphonectria*. Fan et al. [12] reported families and genera of diaporthalean fungi associated with canker and dieback of tree hosts. In addition, dozens of *Diaporthales* taxa were first described in China [13,14,15,16].

Morphological comparisons and phylogenetic analyses have been commonly used to describe the taxa of *Diaporthales* and to confirm their taxonomic placement. Genealogical concordance phylogenetic species recognition (GCPSR) has also been used for the delineation of *Foliocryphiaceae*, *Diaporthe*, *Gnomoniopsis*, and *Coniella* species. GCPSR relies on performing a pairwise homoplasy index coupled with phylogenetic relatedness in a multi-locus dataset and the interpretation of nucleotide differences [17,18].

The present study follows a recently revised classification [1] combined with molecular data, morphology, and pairwise homoplasy index (PHI) test results and introduces seven novel taxa and two newly recorded taxa within the family *Diaporthales* found in Guizhou, China.

## 2. Materials and Methods

### 2.1. Sample Collection and Fungal Strain Isolation

The live plant samples were collected from Wengan, Longli, and Dejiang counties in Guizhou Province, China, in June and September 2021 and March 2022. They were placed in envelopes, taken back to the laboratory, and photographed. Before single-spore isolation, the surfaces of the samples were disinfected by spraying 2 or 3 times with 75% ethanol. The single spore isolation procedure of Chomnunti et al. [19] was followed in order to obtain pure cultures. Each pure culture was spread onto 90 mm diam. Petri dishes containing either potato dextrose agar (PDA) or oatmeal agar (OA) [20]. *Diaporthe* spp. were induced to sporulate by plating them on 2% water agar (WA) [21] containing sterilized pine needles. These dishes were cultured at a constant temperature (25 °C) under a 12 h light/dark regime in a light incubator. Dried holotype specimens were conserved in the Herbarium of the Department of Plant Pathology, Agricultural College, Guizhou University (HGUP). Ex-type cultures were conserved in the Culture Collection at the Department of Plant Pathology, Agriculture College, Guizhou University, China (GUCC).

### 2.2. Morphological Description

The pure cultures were grown on PDA and OA media in a constant-temperature incubator (25 °C) under a 12 h light/dark regime. Culture characteristics were recorded and examined using a stereo microscope (LEICA S9i, Wetzlar, Germany). Morphological observations were made with a Zeiss Scope 5 (Axioscope 5, Shanghai, China) equipped with an AxioCam 208 color camera (ZEN 3.0), and measurements were made with program (ZEN 3.0). Adobe Photoshop CC 2017 was used to make the photoplates. All new taxa were registered in MycoBank [22].

### 2.3. DNA Extraction, PCR Amplification, and Sequencing

Fresh mycelium was scraped from cultures using a sterilized scalpel, and genomic DNA was extracted using Fungal gDNA Kit (Biomiga #GD2416, San Diego, CA, USA) in accordance with the manufacturer’s instructions. Five genes were selected: internal transcribed spacers (ITS), 28S subunit rDNA (LSU), RNA polymerase II subunit 2 (*rpb2*), translation elongation factor 1 (*tef1*), and the β-tubulin gene region (*tub2*). Polymerase chain reactions (PCR) were carried out in 20 μL reaction volume, which contained 10 μL 2 × PCR Master Mix, 7 μL of ddH2O, 1 μL of each primer, and 1 μL of template DNA. The PCR thermal cycle program and primers are shown in Table 1. Purification and sequencing of PCR products were carried out by the Sangon Biotech Company (Shanghai, China). All isolates of all analyzed genes were deposited in GenBank.

### 2.4. Phylogeny

Sequences used in this study (Table 2, Table 3, Table 4 and Table 5) were assembled based on the closest matches from BLASTn search results (https://blast.ncbi.nlm.nih.gov/Blast.cgi) and previous publications [1,29,30,31,32,33,34,35,36,37]. Alignments were conducted with the online version of MAFFT v. 7.505 [38], checked visually, and improved manually where necessary using BioEdit 7.1.3.0 [39]. Sequence matrix v. 1.7.8 [40] was used to concatenate the aligned sequences of the different loci. Ambiguous areas were excluded from the analysis using Aliview [41], and gaps were viewed as missing data. The final alignments were deposited in TreeBASE (www.treebase.org) under accession number 29787.

Phylogenetic analyses were carried out by maximum likelihood (ML), maximum parsimony (MP), and Bayesian inference (BI). The ML analysis was performed using RAxML-HPC BlackBox(8.2.12) [42] partial and general time reversible model (GTR) using the discrete gamma distribution as the evolution model by CIPRES Science Gateway version 3.3 [43]. Non-parametric bootstrap analysis was implemented with 1000 iterations. The resulting duplicates were plotted onto the best-scoring tree previously obtained.

Maximum parsimony (MP) analyses were performed with PAUP on XSEDE (4.a168) on CIPRES Science Gateway v. 3.3 using the heuristic search option with 1000 random sequence addition replicates and tree bisection and reconnection (TBR) with reconnection limit (=8) as the branch-swapping algorithm. Maxtrees were set to 5000 (and not increased). Branches were collapsed, creating polytomies if maximum branch length was zero. Tree length (TL), consistency index (CI), retention index (RI), rescaled consistency index (RC), and homoplasy index (HI) were calculated for each tree generated.

Bayesian inference (BI) analysis was performed by MrBayes 3.2.7a [44] in the CIPRES Science Gateway version 3.3. The optimal substitution model with gamma rates and dirichlet base frequencies for ITS, LSU, *rpb2*, *tef1*, and *tub2* sequences was decided by modelGUI for each locus [45]. The Markov chain Monte Carlo (MCMC) sampling approach was used to calculate posterior probabilities (PP) [46]. Six simultaneous Markov chains were run for 50 million generations and trees were sampled every 1000th generation; thus, 50,000 trees were obtained. The first 25% of trees, representing the burn-in phase of the analyses, were discarded, and the remaining trees were used for calculating posterior probabilities (PP) in the majority rule consensus tree.

The phylogenetic trees were viewed with FigTree v. 1.4.3 [47] and processed with Adobe Illustrator CS5. ML bootstrap support (MLBS) and MP bootstrap support (PBS) equal or greater than 70% [48] and Bayesian posterior probabilities (PP) equal or greater than 0.95 [49] are displayed in the first, second, and third positions on the edited phylogenetic tree, respectively.

### 2.5. Genealogical Phylogenetic Species Recognition (GCPSR) Analysis

Morphologically and phylogenetically related species were analyzed using GCPSR as described by Taylor et al. [17] by the pairwise homogeneity index test (PHI) [50]. The PHI tests were performed in SplitsTree v. 4.17.1 [18,51] as described by Quaedvlieg et al. [52] to determine the level of recombination within phylogenetically closely related species. This test determines the null hypothesis probability (*p*-value) of no recombination within the dataset. When the *p*-value is less than 0.05, we reject the null hypothesis and accept the alternate hypothesis that there is evidence of the presence of recombination. The results were visualized by constructing a split graph using LogDet conversion and Splits options.

## 3. Results

### 3.1. Phylogenetic Analyses

To reveal the phylogenetic position of the family *Foliocryphiaceae*, genera *Diaporthe*, *Gnomoniopsis*, and *Coniella*, within the order *Diaporthales*, phylogenetic analyses were performed with ITS, LSU, *rpb2*, *tef1*, and *tub2* sequence data.

The first sequence dataset of ITS, LSU, *rpb2*, *tef1*, and *tub2* was analyzed to focus on *Cryphonectriaceae* and *Foliocryphiaceae*. The alignment included 43 taxa, including representatives of *Cryphonectriaceae* and *Foliocryphiaceae* and outgroup sequences of *Dwiroopa lythri* (CBS 109755, ex-type strain) and *Dw. punicae* (CBS 143163, ex-type strain) (Table 2). The aligned five-locus datasets comprised 3596 characters of the family *Foliocryphiaceae*, viz. ITS: 1–769, LSU: 770–1617, *rpb2*: 1618–2442, *tef1*: 2443–2952, and *tub2*: 2953–3596. Of these, 2145 characters were constant, 1131 characters were parsimony-informative, and 320 were parsimony-uninformative (gaps were treated as missing). The parameter settings used are shown in Table 6. A RAxML tree was selected to show the topology (Figure 1), and MP and Bayesian analyses resulted in similar topology to ML.

Two new strains of *Chrysofolia coriariae* sp. nov. (GUCC 416.4, ex-type strain and GUCC 416.14) collected during this study in Guizhou Province shared the same branch length with 100% MLBS/99% MPBS/1 PP support and were grouped with the type strains of *Ch. colombiana* (CPC 24986) and *Ch. barringtoniae* (TBRC 5647) with high statistical support, being (76% MLBS/99% MPBS)/(100% MLBS/99% MPBS/1 PP) (Figure 1), respectively. A comparison of the DNA base composition (Table 7) indicated that between our two strains and *Ch. colombiana* (CPC 24986), there were seven different bases in the ITS region, two different bases in the LSU region, and 149 different bases in the *tef1* region. Between GUCC 416.4, GUCC 416.14, and *Ch. barringtoniae* (TBRC 5647), there were 31 different bases in the ITS region and four different bases in the LSU region. Unfortunately, *Ch. colombiana* did not have *rpb2* or *tub2* sequence data, and *Ch. barringtoniae* did not have *rpb2*, *tef1*, or *tub2* sequence data.

Four new strains of *Pseudomastigosporella guizhouensis* sp. nov. (GUCC 406.6, ex-type strain, GUCC 405.3, GUCC 405.4, and GUCC 405.8) from China formed a stable subclade, clustered with the genera *Neocryphonectria*, *Chrysofolia*, and *Foliocryphia*. They formed a well-resolved clade (99% MLBS/94% MPBS/1 PP) within *Foliocryphiaceae*. Our four strains were closer to *N. carpini* (CFCC 53027, ex-type strain) and *N. chinensis* (CFCC 53025, ex-type strain and CFCC 53029) with high support in their respective branches (97% MLBS/1 PP) (Figure 1). A comparison of the DNA bases (Table 7) revealed 116/118/32 base pair differences in ITS, 22/21/21 base pair differences in LSU, no data and 75/76 base pair differences in *rpb2*, and no data and 130/130 base pair differences in *tef1* between our four strains, *N. carpini* (CFCC 53027, ex-type strain), and *N. chinensis* (CFCC 53025, ex-type strain and CFCC 53029) but no *tub2* sequences data for comparison.

The pairwise homoplasy index (PHI) (Figure 2a) test revealed that there was no significant recombination (*p*-value = 1.0) between our strains (GUCC 416.4, GUCC 416.14, GUCC 406.6, GUCC 405.3, GUCC 405.4, and GUCC 405.8) and three other genera in *Foliocryphiaceae* (*Chrysofolia*, *Foliocryphia*, and *Neocryphonectria*).

The second sequence dataset of ITS, *tef1*, and *tub2* was analyzed in combination to infer the interspecific relationships within *Diaporthe*. The alignment included 55 taxa, including the outgroup sequences of *Diaporthella corylina* (CBS 121124, ex-neotype strain) and *Di. cryptica* (CBS 140348, ex-neotype strain) (Table 3). The aligned three-locus datasets comprised 1770 characters of *Diaporthe*, viz. ITS: 1–640, *tef1*: 641–1116 and *tub2*: 1117–1770. Of these, 959 characters were constant, 644 characters were parsimony-informative, and 167 were parsimony-uninformative (gaps were treated as missing). The parameter settings that were used are shown in Table 6. A RAxML tree was selected to show the topology (Figure 3), and MP and Bayesian analyses resulted in similar topology to ML.

Two new strains of *Diaporthe juglandigena* sp.nov. (GUCC 422.16, ex-type strain and GUCC 422.161) from China had a close relationship to *D. chimonanthi* (HGUP191001 and HGUP192087) and *D. caryae* (CFCC 52563, ex-epitype strain and PSCG520), supported by MLBS (96%), PBS (88%), PP (1) and MLBS (100%), PBS (100%), and PP (1), respectively (Figure 3). A comparison of the DNA bases (Table 7) showed that our strains kept some distinction from *D. chimonanthi* (HGUP191001 and HGUP192087) and *D. caryae* (CFCC 52563, ex-type strain and PSCG520) with 18/23 and 2/4 base pair differences in ITS, no data and 6/6 base pair differences in *tef1*, and 5/2 and 17/30 base pair differences in *tub2*. The pairwise homoplasy index (PHI) test (Figure 2b) revealed that there was no significant recombination (*p*-value = 0.5412) among these two strains (GUCC 422.16 and GUCC 422.161) to *D. chimonanthi* (HGUP191001 and HGUP192087) and *D. caryae* (CFCC 52563, ex-epitype strain and PSCG520).

Strains of *Diaporthe eucommiigena* sp.nov. (GUCC 420.9, ex-type strain and GUCC 420.19) shared the same branch length with 100% MLBS/100% MPBS/1 PP support and were grouped with the ex-type strains of *D. passiflorae* (CBS 132527, ex-type strain) and *D. malorum* (CAA734, ex-type strain, CAA740, and CAA752) with high statistical support (100% MLBS/100% MPBS/1 PP) (Figure 3). A comparison of the DNA bases (Table 7) revealed 11/15 bp differences in ITS, 23/19 bp differences in *tef1*, and 13/13 bp differences in *tub2* between the two strains and *D. passiflorae* and *D. malorum*. The PHI test (Figure 2c) did not find statistically significant evidence of recombination (*p*-value = 1.0) between the strains (GUCC 420.9 and GUCC 420.19) and related taxa *D. passiflorae* (CBS 132527, ex-type strain) and *D. malorum* (CAA734, ex-type strain, CAA740, and CAA752).

Strains of *Diaporthe dejiangensis* sp.nov. (GUCC 421.2, ex-type strain and GUCC 421.21) have a close relationship with *D. eres* (CBS 138594 ex-type strain, CAA801) with high support (98% ML, 93% MP, 1 PP) and formed a well-resolved clade sister to *D. eres*. A comparison of the DNA bases (Table 7) showed 4/3/7 bp differences in ITS, 6/7/5 bp differences in *tef1,* and 6/7/13 bp differences in *tub2* between GUCC 421.2 and GUCC 421.21 and the other three strains in the clade. Based on the PHI test (Figure 2d), there was no significant recombination (*p*-value = 1.0) between our strains (GUCC 421.2 and GUCC 421.21) and the sister taxon *D. eres* (CBS 138594, ex-type strain and CAA801).

Strains of *Diaporthe tongrensis* sp.nov. (GUCC 421.10, ex-type strain and GUCC 421.101) formed a highly supported subclade (83% ML, 0.99 PP) with *D. phragmitis* (CBS 138897, ex-type strain). There were eight base pair differences in the ITS and 13 base pair difference in the *tub2*. Unfortunately, *D. phragmitis* (CBS 138897, ex-type strain) did not have *tef1* sequences data for comparison. The PHI test (Figure 2e) did not find statistically significant evidence of recombination (*p*-value = 1.0) between our *Diaporthe* strains (GUCC 421.10, ex-type strain and GUCC 421.101) and related taxa *D. ellipicola* (CGMCC 3.17084, ex-type strain and CGMCC 3.17085) and *D. phragmitis* (CBS 138897, ex-type strain).

Two new strains of *Diaporthe hyperici* sp. nov. (GUCC 414.4, ex-type strain and GUCC 414.41) formed a high-support subclade (100% ML, 100% MP, 1.00 PP) with *D. caulivora* (CBS 127268, ex-type strain, Dip1, and Dpc11). There were 17 base pair differences in ITS, 30 base pair differences in *tef1*, and 11 base pair differences in the *tub2* from our strains based on a DNA base comparison (Table 7). The PHI test (Figure 2f) did not find any statistically significant evidence of recombination (*p*-value = 1.0) between our two strains (GUCC 414.4, GUCC 414.41) and strains of *D. caulivora*.

The third sequence dataset of ITS, LSU, *rpb2*, *tef1*, and *tub2* was analyzed in combination to infer the interspecific relationships within *Gnomoniopsis*. The alignment included 29 taxa, including the outgroup sequences of *Sirococcus tsugae* (CBS 119626) (Table 4). The aligned five-locus datasets comprised 3354 characters of *Gnomoniopsis*, viz. ITS: 1–572, LSU: 573–1423, *rpb2*: 1424–2460, *tef1*: 2461–2861, and *tub2*: 2862–3354. Of these, 2431 characters were constant, 615 characters were parsimony-informative, and 308 were parsimony-uninformative (gaps were treated as missing). The parameter settings that were used are shown in Table 6. A RAxML tree was selected to show the topology (Figure 4), and MP and Bayesian analyses resulted in similar topology to RAxML.

Our strains of *Gnomoniopsis rosae* (GUCC 408.7 and GUCC 408.17) clustered in the same subclade (100% ML, 100% MP, 1.00 PP) with *G. rosae* (CBS 145085), and there were identical sequences in the ITS, LSU, and *rpb2* regions. The phi test results (*p*-value = 1.0) (Figure 2g) of our strains (GUCC 408.7 and GUCC 408.17), *G. rosae* (CBS 145085), *G. angolensis* (CBS 145057), and *G. clavulata* (AR 4313) showed no statistically significant recombination.

The fourth sequence dataset of ITS, LSU, and *tef1* was analyzed in combination to infer the interspecific relationships within *Coniella*. The alignment included 32 taxa, including the outgroup sequences of *C. fragariae* (CBS 172.49, ex-type strain) and *C. nigra* (CBS 165.60, ex-type strain) (Table 5). The aligned three-locus datasets comprised 2165 characters of *Coniella*, viz. ITS: 1–595, LSU: 596–1767, and *tef1*: 1768–2165. Of these, 1717 characters were constant, 385 characters were parsimony-informative, and 63 were parsimony-uninformative (gaps were treated as missing). The parameter settings that were used are shown in Table 6. A RAxML tree was selected to show the topology (Figure 4), and MP and Bayesian analyses resulted in similar topology to RAxML.

Strains of *Coniella quercicola* (GUCC 414.2, GUCC 414.21, GUCC 412.3, GUCC 405.6, and GUCC 405.16) clustered very close to *C. quercicola* (CBS 904.69, ex-type strain, CBS 283.76, and CPC 12133) with only MP support (95% MPBS) (Figure 4). A comparison of the DNA base composition (Table 7) indicated that between *C. quercicola* (CBS 904.69, ex-type strain, CBS 283.76, and CPC 12133) and our five strains (GUCC 414.2, GUCC 414.21, GUCC 412.3, GUCC 405.6, and GUCC 405.16), there were identical sequences in the ITS and LSU regions, but 1/5/10/10/18/7/6 bases were different in the *tef1* region. The PHI test (Figure 2h) did not find statistically significant evidence of recombination (*p*-value = 0.2264) between our five strains and related taxa *C. quercicola* (CBS 904.69, ex-type strain, CBS 283.76, and CPC 12133).

### 3.2. Taxonomy

***Pseudomastigosporella*** S.Y. Wang, Yong Wang bis, and Y. Li, gen. nov.

MycoBank Number: MB846026

Etymology: In reference to *Mastigosporella*, to which this genus is morphologically similar.

Classification: *Foliocryphiaceae*, *Diaporthales*, *Sordariomycetes*.

Description: **Life style**: Parasitic, leaves of *Hypericum patulum* and *Acer palmatum*. **Asexual morph**: *Conidiomata* pycnidial, globose or subglobose, base immersed, separate to aggregated, *mycelium* superficial, fluffy, granular, white or gray-white to pale yellow, exuding light brown-orange to medium brown-orange to deep brown-orange conidial masses, bright yellow or light orange in lactic acid, 2–5 wall layers of olive to gray-green textura angularis. *Conidiophores* reduced to conidiogenous cells. *Conidiogenous cells* arising from base, central cushion of hyaline cells, densely aggregated, slightly thicker, cylindrical to ampulliform, simple, lining the inner cavity of base, mostly hyaline, sometimes pale olive, smooth, cylindrical to ampulliform, straight to curved, sometimes wider at the base. *Conidia* solitary, hyaline, smooth, guttulate, fusoid to ellipsoidal, sometimes long bubble-shaped, straight to curved, aseptate, base tapering with flattened scar, apex with 1 tubular appendage. **Sexual morph**: Unknown.

Type species: *Pseudomastigosporella guizhouensis* S.Y. Wang, Yong Wang bis & Y. Li.

Notes: In *Foliocryphiaceae*, the important morphological characters of asexual morph was to produce dimorphic conidia. The microconidia were minute, cylindrical, aseptate, hyaline to pale brown; macroconidia were fusoid, aseptate, hyaline [1]. However, *Pseudomastigosporella* only had macroconidia but like species in *Mastigosporellaceae* with an apical appendage developing as continuation of conidium body. This feature contradicted the root of “key to genera in *Cryphonectriaceae*, *Foliocryphiaceae*, and *Mastigosporellaceae*” provided by Jiang et al. [1]. However, following our phylogenetic analyses we still proposed that *Pseudomastigosporella* should be placed in *Foliocryphiaceae* family.

***Pseudomastigosporella guizhouensis*** S.Y. Wang, Yong Wang bis, and Y. Li, sp. nov.

MycoBank Number: MB846027, Figure 5.

Etymology: In reference to the location where the fungus was found, being isolated from Guizhou Province.

Type: China, Guizhou Province, Wengan County, on leaves of *Hypericum patulum* and *Acer palmatum*, June 2021, S.Y. Wang (HGUP 406, holotype; HGUP 405, ex-type living culture GUCC 406.6).

Description: **Life style**: Parasitic, leaves of *Hypericum patulum* and *Acer palmatum*. **Asexual morph**: *Conidiomata* pycnidial, globose or subglobose, base immersed, separate to aggregated, *mycelium* superficial, fluffy, granular, white or gray-white to pale yellow, producing light brown-orange to medium brown-orange to deep brown-orange conidial masses, up to 570 μm diam., bright yellow or light orange in lactic acid, 2–5 wall layers of olive to gray-green textura angularis, 50–570 µm diam. *Conidiophores* reduced to conidiogenous cells. *Conidiogenous cells* arising from base, central cushion of hyaline cells, densely aggregated, slightly thicker, cylindrical to ampulliform, simple, lining the inner cavity of base, mostly hyaline, sometimes pale olive, smooth, cylindrical to ampulliform, straight to curved, sometimes wider at the base, 5–20 × 1.5–4.5 µm ($\bar{x}$ = 12 × 3 µm; n = 20). *Conidia* solitary, hyaline, smooth, guttulate, fusoid to ellipsoidal or fish-shaped, sometimes long bubble-shaped, straight to curved, aseptate, 15–31 × (3.5–)5.5–8.5 µm ($\bar{x}$ = 25 × 6.5 µm; n = 30), base tapering with flattened scar, 1.5–4 µm diam., with 1 apical, tubular appendage, 3.5–14.5 µm long. **Sexual morph**: Unknown.

Material examined: China, Guizhou Province, Wengan County, on leaves of *Hypericum patulum* and *Acer palmatum*, June 2021, S.Y. Wang (HGUP 406, holotype; HGUP 405); culture ex-type GUCC 406.6, additional living culture: GUCC 405.3, GUCC 405.4, GUCC 405.8.

Culture characteristics: *Colonies* covering 9 cm Petri dish after 2 weeks at 25 °C and under a 12 h light/dark regime. On PDA, white or gray-white, fluffy, granular, effuse surface, reverse white or beige; on OA, white or gray-white to pale yellow, fluffy, granular, effuse surface, exuding light brown-orange to medium brown-orange to deep brown-orange conidial masses, reverse white or beige to pale yellow.

Notes: Although *Ps. guizhouensis* produces macroconidia with one tubular apical appendage, the conidia of this species differs in shape and size from those of *Chrysofolia* and *Foliocryphia.* The conidia of *Chrysofolia* are ellipsoidal and measure (4–)6–7.5(–10) × (2–)2.5(–3) µm [53], while those of *Foliocryphia* are also ellipsoidal and measure (5–)6–8(–9) × (2–)2.5(–3) µm [54]. The results of the DNA base comparisons (Table 7) showed that there were striking differences in each gene among our four *Pseudomastigosporella* strains and adjacent genera. Based on its distinct morphological characteristics, DNA phylogeny, DNA base differences, and pairwise homoplasy index (PHI) test results, *Pseudomastigosporella* was described here as a new genus in *Foliocryphiaceae* with *Ps. guizhouensis* as the type species.

***Chrysofolia coriariae*** S.Y. Wang, Yong Wang bis, and Y. Li, sp. nov.

MycoBank Number: MB845958, Figure 6.

Etymology: In reference to the host plant *Coriaria nepalensis*, from which this fungus was collected.

Type: China, Guizhou Province, Longli County, on leaves of *Coriaria nepalensis*, June 2021, S.Y. Wang (HGUP 416, holotype; ex-type living culture GUCC 416.4).

Description: **Life style**: Parasitic, leaves of *Coriaria nepalensis*. **Asexual morph**: *Conidiomata* pycnidial, globose or subglobose, separate to aggregated, *mycelium* superficial and immersed, exuding yellow to bright orange to brown-orange wet conidial masses, green-brown in lactic acid, but bright yellow or light orange in sterile water, 2–6 wall layers of green-brown to brown textura angularis, 50–400 µm diam.; neck 15–60 µm long, 50–200 µm diam. where attached to the globose, terminating in an obtusely rounded apex. *Conidiophores* reduced to conidiogenous cells. *Conidiogenous cells* arising from base, central cushion of hyaline cells, densely aggregated, slightly thicker, tapering or cylindrical to ampulliform, simple, lining the inner cavity of base, hyaline, smooth, cylindrical to ampulliform, straight to curved, wider at the base, 5–20 × 1–3.5 µm. *Conidia* solitary, hyaline, smooth, ellipsoidal, or crescent-shaped, straight to allantoid, apex obtuse, base tapering with flattened scar, 0.5 µm diam., 5.5–9 × 2–4 µm ($\bar{x}$ = 7 × 2.7 µm; n = 30). **Sexual morph**: Unknown.

Material examined: China, Guizhou Province, Longli County, on leaves of *Coriaria nepalensis*, June 2021, S.Y. Wang (HGUP 416, holotype); culture ex-type GUCC 416.4, additional living culture: GUCC 416.14.

Culture characteristics: *Colonies* culturing under a controlled temperature light incubator at 25 °C and under a 12 h light/dark regime for 2 weeks. *Colonies* on PDA 75–90 mm diam. after 2 weeks at 25 °C, light brown to white or gray-white, felty, effuse surface, with white fluffy even mycelium margin, reverse brown to light brown to white edge. *Colonies* on OA 65–85 mm diam. after 2 weeks at 25 °C, light brown to white or gray-white, flat surface, exuding orange or brown conidial masses, reverse light brown to white or gray-white.

Notes: The conidiogenous cells of *Ch. coriariae* (5–20 × 1–3.5 µm) are longer than those of *Ch. colombina* (5–8 × 2–3 µm) [53] and *Ch. barringtoniae* (3–7.5 × 2–3 µm) [55], and the conidiomata of *Ch. coriariae* (5–400 µm) are larger than those of *Ch. barringtoniae* (5–110 µm) [55]. *Chrysofolia coriariae* was phylogenetically distinct from other known species (Figure 1) and displayed some differences in the DNA base comparison with *Ch. colombina* and *Ch. barringtoniae* (Table 7). The new isolates were described here as a new species based on their distinct morphological characteristics, DNA phylogeny, DNA base differences, and pairwise homoplasy index (PHI) test results.

***Diaporthe juglandigena*** S.Y. Wang, Yong Wang bis, and Y. Li, sp. nov.

MycoBank Number: MB845959, Figure 7.

Etymology: Name refers to the plant host genus (*Juglans*) from which this fungus was collected.

Type: China, Guizhou Province, Dejiang County, on the peel of *Juglans regia*, September 2021, S.Y. Wang (HGUP 422, holotype; ex-type living culture GUCC 422.16).

Description: **Life style**: Parasitic, peels of *Juglans regia*. **Asexual morph**: *Conidiomata* pycnidial, scattered, immersed or superficial, irregular globose and subglobose to slightly erumpent, black conidial masses surrounded by white mycelium, up to 2 mm diam, exuding transparent drops of water. *Conidiophores* reduced to conidiogenous cells. *Conidiogenous cells* densely aggregated, slightly thicker, subulate, simple, rarely branched above, simple, tapering, hyaline, smooth, 19–34 × 1–2.5 µm ($\bar{x}$ = 27 × 1.7 µm; n = 20), wider at base, rarely branched, densely aggregated, cylindrical, straight to sinuous. *Alpha conidia* hyaline, fusoid to ellipsoidal, asymmetrical, frequently guttulate, smooth-walled, 0–1-septate, tapering towards both ends, mostly straight, 5–8 × 2–3 µm ($\bar{x}$ = 6.4 × 2.3 µm; n = 30). *Beta conidia* infrequent, hyaline, filiform, aseptate, smooth, eguttulate, apex acute, mostly curved, 23–36 × 1–2 µm ($\bar{x}$ = 31 × 1.3 µm; n = 10). *Gamma conidia* not observed. **Sexual morph:** Not observed.

Material examined: China, Guizhou Province, Dejiang County, on the peel of *Juglans regia*, September 2021, S.Y. Wang (HGUP 422, holotype); culture ex-type GUCC 422.16, additional living culture: GUCC 422.161.

Culture characteristics: *Colonies* covering 9 cm Petri dish after 2 weeks at 25 °C and under a 12 h light/dark regime; spreading with uneven aerial *mycelium*. On PDA, surface with abundant aerial mycelium, white or gray-white to pale brown; reverse white to pale yellow to light brown. On OA surface with white or pale white to pale yellow thin aerial mycelium, with black conidial masses surrounded by white or gray-white mycelium; reverse white or beige to light yellow. On pine needles with irregular dark green to black subglobose conidial masses surrounded by thick white mycelium.

Notes: The conidiomata of *D. juglandigena* (2 mm diam.) are larger than those of *D. chimonanthi* (= *Phomopsis chimonanthi*) (150–238 µm wide, 130–230 µm high) [56] and *D. caryae* (310–325 µm diam.) [11]. The conidiogenous cells and beta conidia of *D. juglandigena* (19–34 × 1–2.5 µm, 23–36 × 1–2 µm) are longer than those of *Phomopsis chimonanthi* (13–25 × 1.6–2.5 µm, 15–18 × 1–1.5 µm) and *D. caryae* (7–11 × 1.4–2.2 µm, 15–34 × 1.1–1.4 µm), while the alpha conidia of *D. juglandigena* (5–8 × 2–3 µm) are shorter than those of *P. chimonanthi* (6.6–8.8 × 1.8–2.2 µm) and *D. caryae* (7–8.5 × 2.1–2.5 µm). *Diaporthe juglandigena* was phylogenetically distinct from the species presently known from DNA analyses (Figure 3). The results of DNA base comparisons (Table 7) showed that there were significant differences in three loci between our two *D. juglandigena* strains and sister species (*D. chimonanthi* and *D. caryae*). Based on its distinct morphological characteristics, DNA phylogeny, DNA base differences, and pairwise homoplasy index (PHI) test results, *D. juglandigena* was described here as a new species.

***Diaporthe eucommiigena*** S.Y. Wang, Yong Wang bis, and Y. Li, sp. nov.

MycoBank Number: MB845961, Figure 8.

Etymology: eucommiigena, in reference to plant host (*Eucommia ulmoides*)*,* from which the fungus was isolated.

Type: China, Guizhou Province, Guiyang, Huaxi District, South Campus of Guizhou University, on dead woods of *Eucommia ulmoides*, March 2022, S.Y. Wang (HGUP 420, holotype; ex-type living culture GUCC 420.9).

Description: **Life style**: Saprobic, dead woods of *Eucommia ulmoides*. **Asexual morph**: *Conidiomata* pycnidial, separated but sometimes aggregated, immersed, sometimes superficial, irregular globose or subglobose, forming dark olive or dull green to black conidial masses, up to 2 mm diam., sometimes surrounded by white mycelium. *Conidiophores* reduced to conidiogenous cells. *Conidiogenous cells* densely aggregated, slightly thicker, subulate, hyaline, simple, rarely branched above, simple, hyaline, smooth, 12–27.5 × 1.5–3 µm ($\bar{x}$ = 19 × 2.2 µm; n = 20), wider at base, tapering at apex, rarely branched, densely aggregated, cylindrical, slightly bent. *Alpha conidia* hyaline, fusoid to ellipsoidal, frequently guttulate, asymmetrical, smooth-walled, 0–1-septate, tapering towards both ends, mostly straight, 5.5–8 × 1.5–3 µm ($\bar{x}$ = 7 × 2.3 µm; n = 30). *Beta conidia* hyaline, filiform, aseptate, smooth, eguttulate, apex acute, mostly curved, 27–37 × 1–2 µm ($\bar{x}$ = 32 × 1.3 µm; n = 10). *Gamma conidia* hyaline, fusoid to ellipsoidal, frequently guttulate, smooth, aseptate, straight, tapering at apex, 7.5–10 × 1.5–2.5 µm ($\bar{x}$ = 8.6 × 2.1 µm; n = 20). **Sexual morph**: Not observed.

Material examined: China, Guizhou Province, Guiyang, Huaxi District, South Campus of Guizhou University, on dead wood of *Eucommia ulmoides*, March 2022, S.Y. Wang (HGUP 420, holotype); culture ex-type GUCC 420.9, additional living culture: GUCC 420.19.

Culture characteristics: *Colonies* covering 9 cm Petri dish after 2 weeks at 25 °C and under a 12 h light/dark regime; spreading with uneven aerial *mycelium*. On PDA, surface with abundant white to pale yellow, uneven zonated aerial mycelium and margin, distinctly imbricated like a flower; reverse with pale yellow to light brown and pale pink, uneven zonated aerial mycelium and margin, exuding abundant dark green to black spots with age. On OA surface with uneven white to olive aerial mycelium, forming black conidial masses surrounded by white or gray-white mycelium; reverse white to olive, irregular. On pine needles with irregular dark green to black subglobose conidial masses surrounded by thick gray-white mycelium.

Notes: The conidiomata of *Diaporthe eucommiigena* (2 mm diam.) are larger than those of *D. passiflorae* (300 µm diam.) [57]. The conidiogenous cells of *D. passiflorae* are 2–3-septate, while those of *D. eucommiigena* are aseptate. The conidiogenous cells of *D. eucommiigena* (12–27.5 × 1.5–3 µm) are smaller than those of *D. passiflorae* (20–30 × 2.5–4 µm), and the beta conidia of *D. eucommiigena* (27–37 × 1–2 µm) are longer than those of *D. passiflorae* (14–20 × 1.5–2 µm) and *D. malorum* (17.4–26.6 × 0.8–2 µm) [21]. Gamma conidia were not observed in *D. passiflorae* but are present in *D. eucommiigena*. *Diaporthe eucommiigena* was phylogenetically distinct from the species presently known based on the DNA data (Figure 3). A comparison of the DNA bases (Table 7) showed significant difference between *D. eucommiigena* and adjacent species (*D. passiflorae* and *D. malorum*). Based on its distinct morphological characteristics, DNA phylogeny, DNA base differences, and pairwise homoplasy index (PHI) test results, *D. eucommiigena* was described here as a new species.

***Diaporthe dejiangensis*** S.Y. Wang, Yong Wang bis, and Y. Li, sp. nov.

MycoBank Number: MB845962, Figure 9.

Etymology: Name refers to the location (Dejing), from where the host plant was collected.

Type: China, Guizhou Province: Dejiang County, on the peel of *Juglans regia*, September 2021, S.Y. Wang (HGUP 421, holotype; ex-type living culture GUCC 421.2).

Description: **Life style**: Parasitic, peels of *Juglans regia*. **Asexual morph**: *Conidiomata* pycnidial, solitary to aggregated, immersed or superficial, irregular globose or subglobose to depressed, exuding white to dark brown to black conidial masses, with age surrounded by thin white mycelium, up to 2 mm diam., 4–7 wall layers of olive textura angularis. *Conidiophores* reduced to conidiogenous cells. *Conidiogenous cells* densely aggregated, hyaline, smooth, cylindrical, wider at base, mostly straight, phialidic, simple, subcylindrical, tapering towards apex, hyaline, smooth, 9.5–17 × 1–3 µm ($\bar{x}$ = 13 × 1.8 µm; n = 30), mostly straight, rarely branched. *Alpha conidia* hyaline, fusiform to ellipsoidal, frequently guttulate, asymmetrical, smooth-walled, 0–1-septate, rounded towards both ends, mostly straight, 6–8.5 × 1.5–3 µm ($\bar{x}$ = 7 × 2.3 µm; n = 30). *Beta conidia* and *gamma conidia* not observed. **Sexual morph**: Not observed.

Material examined: China, Guizhou Province, Dejiang County, on the peel of *Juglans regia*, September 2021, S.Y. Wang (HGUP 421, holotype); culture ex-type GUCC 421.2, additional living culture: GUCC 421.21.

Culture characteristics: *Colonies* covering 9 cm diam. Petri dish after 2 weeks at 25 °C under a 12 h light/dark regime. On PDA surface with thick aerial mycelium, flat, velvet, white and beige; reverse white to pale yellow. On OA surface with white or pale white thin aerial mycelium, exuding black conidial masses, surrounded by white mycelium; reverse white or beige. On pine needles, irregular, black, globose conidial masses surrounded by thick white mycelium.

Notes: The conidiomata of *D. dejiangensis* (2 mm diam.) are larger than those of *D. cotoneastri* (1.5 mm diam.) [58], while its alpha conidia (6–8.5 × 1.5–3 µm) are smaller than those of *D. cotoneastri* (6–10 × 2–3 µm). Neither beta nor gamma conidia were observed for *D. dejiangensis,* while *D. cotoneastri* produced beta conidia (18–25 × 1 µm). *Diaporthe dejiangensis* was phylogenetically distinct from the species presently known based on the DNA data (Figure 3). The results of the DNA base comparisons are shown in Table 7 and indicate that there were many base differences among three genes. Based on its distinct morphological characteristics, DNA phylogeny, DNA base differences, and pairwise homoplasy index (PHI) test results, *D. dejiangensis* was described here as a new species.

***Diaporthe tongrensis*** S.Y. Wang, Yong Wang bis, and Y. Li, sp. nov.

MycoBank Number: MB845963, Figure 10.

Etymology: tongrensis, in reference to the city (Tongren) where the fungus was isolated.

Type: China, Guizhou Province, Tongren City, Dejiang County, on the peel of *Juglans regia*, September 2021, S.Y. Wang (HGUP 421, holotype; ex-type living culture GUCC 421.10).

Description: **Life style**: Parasitic, peels of *Juglans regia*. **Asexual morph**: *Conidiomata* pycnidial, separated, immersed or superficial, irregular globose or subglobose to depressed, exuding black conidial masses surrounded by white mycelium, up to 2.5 mm diam., 7–10 wall layers of olive textura angularis. *Conidiophores* reduced to conidiogenous cells. *Conidiogenous cells* densely aggregated, slightly thicker, subulate, simple, rarely branched above, tapering, hyaline, smooth, 12–24 × 1.5–2.5 µm ($\bar{x}$ = 16 × 1.8 µm; n = 20), wider at base, cylindrical, straight to sinuous. *Alpha conidia* hyaline, fusoid to ellipsoidal, asymmetrical, smooth-walled, 1-septate, rounded towards both ends, mostly straight, 5.5–7.5 × 2–3 µm ($\bar{x}$ = 6.5 × 2.5 µm; n = 30). *Beta conidia* infrequent, hyaline, filiform, aseptate, eguttulate, smooth, apex acute, base slightly truncate, mostly straight, sometimes curved, 20–30 × 1–2 µm ($\bar{x}$ = 25.5 × 1.5 µm; n = 15). *Gamma conidia* not observed. **Sexual morph**: Not observed.

Material examined: China, Guizhou Province, Dejiang County, on the peel of *Juglans regia*, September 2021, S.Y. Wang (HGUP 421, holotype); culture ex-type GUCC 421.10, additional living culture: GUCC 421.101.

Culture characteristics: *Colonies* covering 9 cm diam. Petri dish after 2 weeks at 25 °C and a 12 h light/dark regime; spreading with aerial mycelium and uneven zonation. On PDA, surface with abundant aerial mycelium, with white uneven zonated aerial mycelium in the middle; reverse with white to pale yellow to light brown, uneven zonated aerial mycelium. On OA, surface with white or pale white, thin aerial *mycelium*, exuding black conidial masses surrounded by white mycelium; reverse white or beige. On pine needles, irregular black subglobose conidial masses surrounded by white mycelium.

Notes: The conidiomata of *D. tongrensis* (2.5 mm diam.) are larger than those of *D. phragmitis* (250 µm diam.) [59], and its conidiogenous cells are aseptate, while those of *D. phragmitis* are 1–3-septate. The conidiogenous cells of *D. tongrensis* (12–24 × 1.5–2.5 µm) are smaller than those of *D. phragmitis* (20–30 × 3–4 µm). The alpha conidia of *D. tongrensis* (5.5–7.5 × 2–3) are shorter than those of *D. phragmitis* (7–8 × 2–3 µm). *Diaporthe tongrensis* formed beta conidia, while neither beta nor gamma conidia were observed in *D. phragmitis*. *Diaporthe tongrensis* is phylogenetically distinct from the presently known species based on the DNA data (Figure 3). *D. tongrensis* showed some differences in the DNA base comparison with *D. phragmitis*, as shown in Table 7. *Diaporthe tongrensis* was described as a new taxon based on the high phylogenetic support for the clade, distinct morphological characteristics, DNA phylogeny, DNA base differences, and pairwise homoplasy index (PHI) test results with adjacent species.

***Diaporthe hyperici*** S.Y. Wang, Yong Wang bis, and Y. Li, sp. nov.

MycoBank Number: MB845965, Figure 11.

Etymology: Name refers to *Hypericum patulum*, the host genus from which this fungus was collected.

Type: China, Guizhou Province, Longli County, on leaves of *Hypericum patulum*, June 2021, S.Y. Wang (HGUP 416, holotype; ex-type living culture GUCC 416.4).

Description: **Life style**: Parasitic, leaves of *Hypericum patulum*. **Asexual morph**: *Conidiomata* pycnidial, separated or aggregated, immersed or superficial, globose or subglobose, deep brown to black, exuding black droplets from central ostioles with age, up to 3 mm diam., 6–10 wall layers of brown-green or brown textura angularis. *Conidiophores* reduced to conidiogenous cells. *Conidiogenous cells* densely aggregated, smooth, cylindrical, subulate, straight, phialidic, simple, cylindrical, hyaline, smooth, 11–21 × 1–3 µm ($\bar{x}$ = 17 × 1.9 µm; n = 30), slightly tapered towards apex, apex with inconspicuous periclinal thickening. *Alpha conidia* hyaline, fusoid to ellipsoidal, asymmetrical, smooth-walled, 0–1-septate, tapering towards both ends, mostly straight, 5–9.5 × 1.5–3 µm ($\bar{x}$ = 7 × 2.2 µm; n = 30). *Beta conidia* hyaline, spindle-shaped, aseptate, smooth, apex subacutate, base slightly truncate, sometimes straight, mostly curved, 10–20 × 1–2.5 µm ($\bar{x}$ = 15 × 1.5 µm; n = 30). *Gamma conidia* not observed. **Sexual morph**: Not observed.

Material examined: China, Guizhou Province, Longli County, on leaves of *Hypericum patulum*, June 2021, S.Y. Wang (HGUP 416, holotype); culture ex-type GUCC 416.4, additional living culture: GUCC 416.14.

Culture characteristics: *Colonies* covering 9 cm diam. Petri dish after 2 weeks at 25 °C and a 12 h light/dark regime. On PDA, surface with thick, white uneven zonated aerial mycelium, exuding brown to dark orange conidial masses; reverse white to pale yellow to light brown, uneven zonated aerial *mycelium* and margin. On OA, surface with white or gray-white to pale yellow, fluffy, granular aerial mycelium, exuding white to dark green to black conidial masses; reverse white or beige to pale yellow.

Notes: The conidiomata of *D. hyperici* (3 mm diam.) are larger than those of *D. caulivora* (230–310 µm diam.) [60], but the alpha conidia of *D. hyperici* (5–9.5 × 1.5–3 µm) are shorter than those of *D. caulivora* (8.9–9.2 × 2.4–2.5 µm). *Diaporthe caulivora* produces a sexual morph with unitunicate asci, while *D. hyperici* has no known sexual morph. *Diaporthe hyperici* was phylogenetically distinct from the presently known species based on the DNA data (Figure 3). A comparison of the DNA bases (Table 7) showed significant difference between *D. hyperici* and sister species (*D. caulivora*). Based on its distinct morphological characteristics, DNA phylogeny, DNA base differences, and pairwise homoplasy index (PHI) test results, *D. hyperici* was described here as a new species.

***Gnomoniopsis rosae*** Crous, Persoonia 41: 305 (2018).

MycoBank Number: MB 828203, Figure 12.

Description: **Life style**: Parasitic, leaves of *Rosa* sp. (*Rosaceae*). **Asexual morph**: *Conidiomata* erumpent, separated, immersed or superficial, globose to depressed, initially appearing deep brown to black, slowly oozing transparent white or pale-yellow oily spheres with age, up to 600 µm diam., 5–7 wall layers of olive brown to brown textura angularis. *Conidiophores* reduced to conidiogenous cells. *Conidiogenous cells* lining the inner cavity, hyaline, smooth, subcylindrical, branched at base or not, frequently branched above, simple, tapering, hyaline, smooth, subcylindrical, tapering towards apex, 6–15 × 1–3.5 µm ($\bar{x}$ = 11 × 2 µm; n = 20). *Conidia* solitary, aseptate, fusoid, hyaline, asymmetrical, guttulate, smooth-walled, rounded to acute apex, 6–12 × 2–4 µm ($\bar{x}$ = 8.5 × 3 µm; n = 30). **Sexual morph**: Unknown.

Materials examined: China, Guizhou Province, Wengan County, on leaves of *Rosa* sp. (*Rosaceae*), June 2021, S.Y. Wang (HGUP 408, living cultures GUCC 408.7 and GUCC 408.17) (**new country record**).

Culture characteristics: *Colonies* cultured at 25 °C and a 12 h light/dark regime for 2 weeks on PDA 60–85 mm diam., forming a circle of transparent mycelium in the center, followed by a circle of white or gray-white thick ridges, then uneven zonated aerial mycelium, slightly imbricated, thick, initially appearing white to pale yellow, slowly turning olive-gray with age outside the two concentric rings, with an uneven edge; reverse transparent to white or olive and white uneven imbricated zonated to white or light brown uneven edge. *Colonies* on OA covering the whole dish, pale white or light gray-white, flat surface, exuding deep brown to black conidial masses, slowly oozing transparent white or pale-yellow oily spheres with age, reverse pale white or light gray-white.

Notes: *Gnomoniopsis* represented a genus of mostly host-specific fungi [61,62]. *Gnomoniopsis rosae* (GUCC 408.7 and GUCC 408.17) was phylogenetically identical to the ex-type strain (CBS 145085) isolated by Crous et al. [31] in ITS, LSU, and *rpb2*, and we also supplemented the DNA sequences of this species with *tef1* and *tub2* genes. The DNA base comparison results are shown in Table 7; there were no DNA base differences among several genes. The isolates of *G. rosae* were newly recorded for China based on their morphological characteristics, DNA phylogeny, DNA base differences, and pairwise homoplasy index (PHI) test results.

***Coniella quercicola*** (Oudem.) L.V. Alvarez and Crous, Studies in Mycology 85: 27 (2016).

MycoBank Number: MB 817831, Figure 13.

Description: **Life style**: Parasitic, leaves of *Hypericum patulum*, *Aralia chinensis* and *Acer palmatum*. **Asexual morph**: *Conidiomata* pycnidial, separated, immersed or superficial, globose to depressed, initially appearing hyaline or light yellow becoming deep brown to black with age, up to 300 µm diam., 2–5 wall layers of dark brown textura angularis. *Conidiophores* reduced to conidiogenous cells. *Conidiogenous cells* central cushion of hyaline cells, densely aggregated, slightly thicker, subulate, simple, frequently branched above, tapering, hyaline, smooth, 12–22 × 3–4.5 µm ($\bar{x}$ = 17 × 3.5 µm; n = 15), surrounded by a gelatinous coating, apex with visible periclinal thickening. *Conidia* hyaline, asymmetrical, smooth-walled, cylindrical, slightly curved to naviculate, aseptate, rounded to acute apex, tapered to a subtruncate base, germ slits absent, 10–23 × 2.5–4.5 µm ($\bar{x}$ = 17 × 3.5 µm; n = 30). **Sexual morph**: Unknown.

Materials examined: China, Guizhou Province, Longli and Wengan counties, on leaves of *Hypericum patulum*, *Aralia chinensis* and *Acer palmatum*, June 2021, S.Y. Wang (HGUP 414, HGUP 412 and HGUP 405, living cultures GUCC 414.2, GUCC 414.21, GUCC 412.3, GUCC 405.6 and GUCC 405.16). (**new host and country record**).

Culture characteristics: *Colonies* covering 9 cm diam. Petri dish after 2 weeks at 25 °C and a 12 h light/dark regime; spreading with sparse aerial mycelium and uneven zonation. On PDA, surface with thin, white uneven zonated aerial mycelium, distinctly imbricated like a flower, producing a few pale yellow to dark brown to black conidial masses; reverse with white and pale brown, uneven zonated, imbricated aerial mycelium. On OA, surface with white or gray-white to pale yellow, fluffy, granular aerial mycelium, producing pale yellow to dark brown to black conidial masses from the center; reverse white or beige to pale yellow.

Notes: *Coniella quercicola* was originally described as *Macroplodia quercicola* on the leaves of *Quercus robur* collected in the Netherlands. It was described as having pale-brown, cylindrical conidia, 24 × 4 µm [63]. Our strains (GUCC 414.2, GUCC 414.21, GUCC 412.3, GUCC 405.6, and GUCC 405.16) are phylogenetically identical to the neotype culture (CBS 904.69) of *C. quercicola* isolated and designated by Alvarez et al. [64] in ITS, LSU, and *tef1* genes (Figure 14). According to the results of the DNA base comparison (Table 7), we note that base differences almost only occur in the *tef1* region. The identification of *C. quercicola* was based on its morphological characteristics, DNA phylogeny, DNA base differences, and pairwise homoplasy index (PHI) test results.

## 4. Discussion and Conclusions

Families, genera, and species within *Diaporthales* are now characterized and separated based on a combination of morphology and molecular data [12,29,30,64,65,66,67,68,69,70,71,72]. The present study described and illustrated nine species (within five genera) of *Diaporthales* isolated from various host plants in Guizhou Province, China, including *Gnomoniopsis* mostly as host-specific fungi [61,62,65,73]. Based on their unique morphological characteristics, DNA phylogeny, DNA base differences, and pairwise homoplasy index (PHI) test evaluations, we described one new genus, seven new species, and two new fungal records for China. Only asexual morphology was observed for all the taxa described in this paper.

*Foliocryphiaceae* (*Diaporthales*) was established by Jiang et al. [1] based on the type genus *Foliocryphia* [54] and two allied genera, *Chrysofolia* [53] and *Neocryphonectria* [1]. *Chrysofolia* and *Foliocryphia* were originally placed in the family *Cryphonectriaceae* but they were transferred to *Foliocryphiaceae* by Jiang et al. [1]. Species of *Chrysofolia* usually exude a yellow slimy mass of conidia from a globose pycnidium with an immersed base. Only two species are listed in MycoBank (www.mycobank.org; accessed on 8 October 2022), *Ch. colombiana* [53], a pathogen of *Eucalyptus urophylla* from Colombia, and *Ch. barringtoniae* [55], an endophyte of *Barringtonia acutangula* from Thailand. *Chrysofolia coriariae* sp. nov. observed in the present study represents the first taxon of *Chrysofolia* in China.

*Diaporthe* is a large genus in *Diaporthaceae* with 1168 epithets listed in Index Fungorum (http://www.indexfungorum.org/; accessed on 4 July 2022) but only one-fifth of these taxa have been studied with molecular data [73,74,75]. The sexual morph of *Diaporthe* is characterized by immersed perithecial ascomata and an erumpent pseudostroma with more or less elongated perithecial necks; unitunicate clavate to cylindrical asci; and fusoid, ellipsoid to cylindrical, septate or aseptate, hyaline ascospores, which are biseriately to uniseriately arranged in the ascus, sometimes having appendages [29,30,76]. The asexual morph is characterized by ostiolate conidiomata, with cylindrical phialides producing three types of hyaline, aseptate conidia [76,77]. Type I α-conidia are hyaline, fusiform, straight, guttulate, or eguttulate; aseptate; and smooth-walled. Type II β-conidia are hyaline, filiform, straight or hamate, aseptate, smooth-walled, and eguttulate. Type III γ-conidia are rarely produced, and are hyaline, multiguttulate, and fusiform to subcylindrical with an acute or rounded apex, while the bases are sometimes truncate. Five new taxa of *Diaporthe* were introduced, which indicates that more potential novel and known taxa in this genus could be discovered because of the rich biodiversity in Guizhou Province.

*Gnomoniaceae* is a large family within *Diaporthales*, containing 38 accepted genera [65,78,79,80,81]. Among them, *Gnomoniopsis* is a well-delimited genus inhabiting the leaves, branches, and fruits of hosts in three families: *Fagaceae*, *Onagraceae*, and *Rosaceae* [62,65,73]. The sexual morph of *Gnomoniaceae* is characterized by ascomata that are generally immersed, solitary, or aggregated in an undeveloped stroma [6,61]. The perithecia are dark brown to black and pseudoparenchymatous with central, eccentric, or lateral necks [6,61]. The asci usually have an inconspicuous or distinct apical ring. Ascospores are generally small, hyaline, and uniseptate. The asexual morph is characterized by acervular or pycnidial conidiomata, phialidic conidiogenous cells, and non-septate conidia [82]. *Gnomoniopsis rosae* in our study was isolated as asexual morph from *Rosa* sp. and was newly recorded for China.

The family *Schizoparmeaceae* (*Diaporthales*) was introduced by Rossman et al. [6]. Historically, the family consisted of three genera, two of which only produce asexual morphs (*Coniella* and *Pilidiella*), while one can produce sexual morphs (*Schizparme*) [6]. This family was reassessed by Alvarez et al. [64], who proposed that *Pilidiella* is a taxonomic synonym of *Coniella*. *Coniella* was erected by Höhnel [83] and typified by *C. pulchella*, [84] who separated the genus into *Euconiella* (with dark conidia) and *Pseudoconiella* (with pale conidia) [64]. The key characteristics of *Coniella* are erumpent, brown, or black ascomata or conidiomata that later become superficial and an irregularly thickened peridium with plate-like ornamentation and one-celled ascospores, initially hyaline and later becoming pale to dark brown [30]. The present isolates of *C. quercicola* represent a new country record for China and new host records.

The molecular data provided evidence that our new genus belongs to *Foliocryphiaceae*, although in morphology it is similar to *Mastigosporella* in *Mastigosporellaceae*. In this molecular era, morphological conclusions are increasingly being reduced to a subordinate or even insignificant position. Thus, we accepted the phylogenetic conclusion to create the monotypic genus *Pseudomastigosporella*. Despite this, we still require additional strains of *Diaporthales* in order to compare the genome-wide information of members in this order due to the high level of similarities in morphology but measurable differences in molecular data. Currently, there are too few data for the adequate comparison of fungi.

## Figures and Tables

**Figure 1 jof-08-01301-f001:**
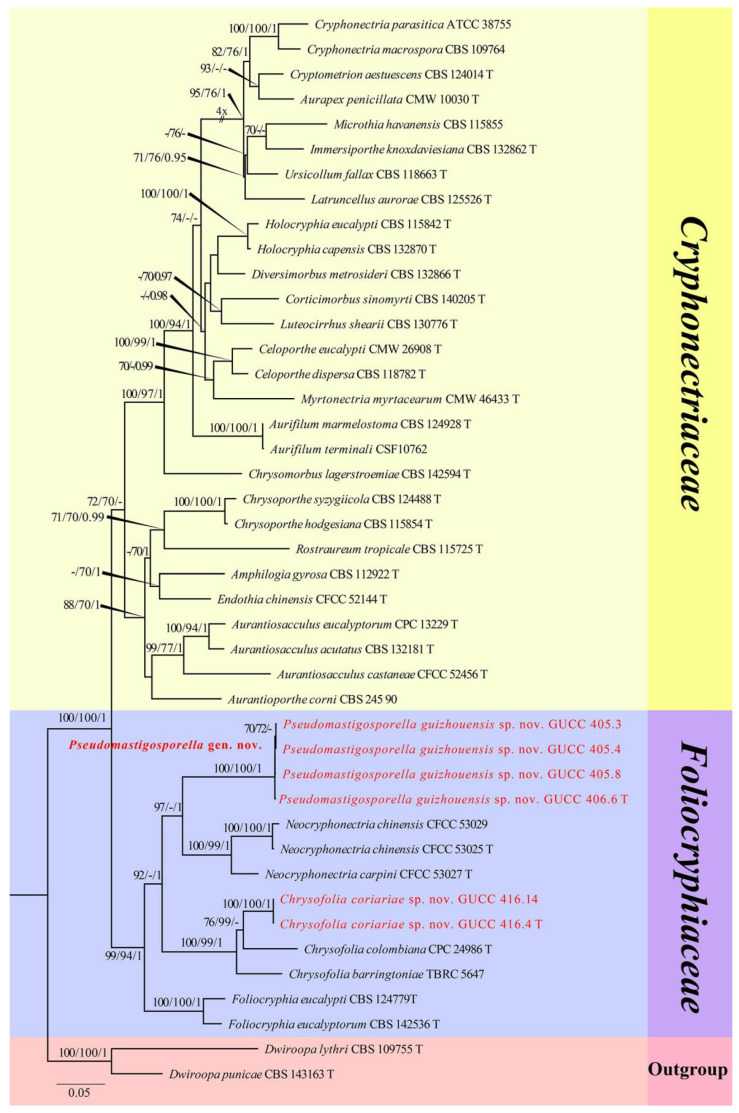
Phylogram generated from RAxML analysis of a concatenated ITS-LSU-*rpb2*-*tef1*-*tub2* sequence dataset to represent the phylogenetic relationships of taxa in *Foliocryphiaceae* and *Cryphonectriaceae*. The tree was rooted with *Dwiroopa lythri* (CBS 109755, ex-type strain) and *Dw. punicae* (CBS 143163, ex-type strain). Bootstrap support values for ML and MP equal to or greater than 70% and Bayesian posterior probabilities equal to or higher than 0.95 PP are indicated above the nodes as ML/MP/PP. Support values lower than 70% ML/MP and 0.95 PP are indicated by a hyphen (-). The newly generated sequences are indicated in red.

**Figure 2 jof-08-01301-f002:**
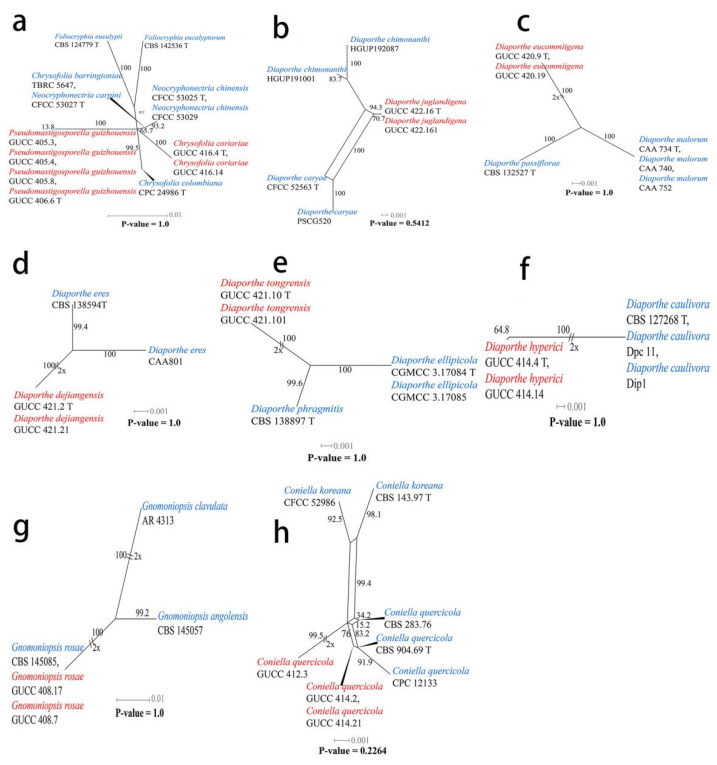
Results of the pairwise homoplasy index (PHI) test of closely related species using both LogDet transformation and splits decomposition. Our strains are indicated in red, other involved strains are indicated in blue. (**a**) *Foliocryphiaceae*. (**b**–**f**) *Diaporthe*. (**g**) *Gnomoniopsis*. (**h**) *Coniella*.

**Figure 3 jof-08-01301-f003:**
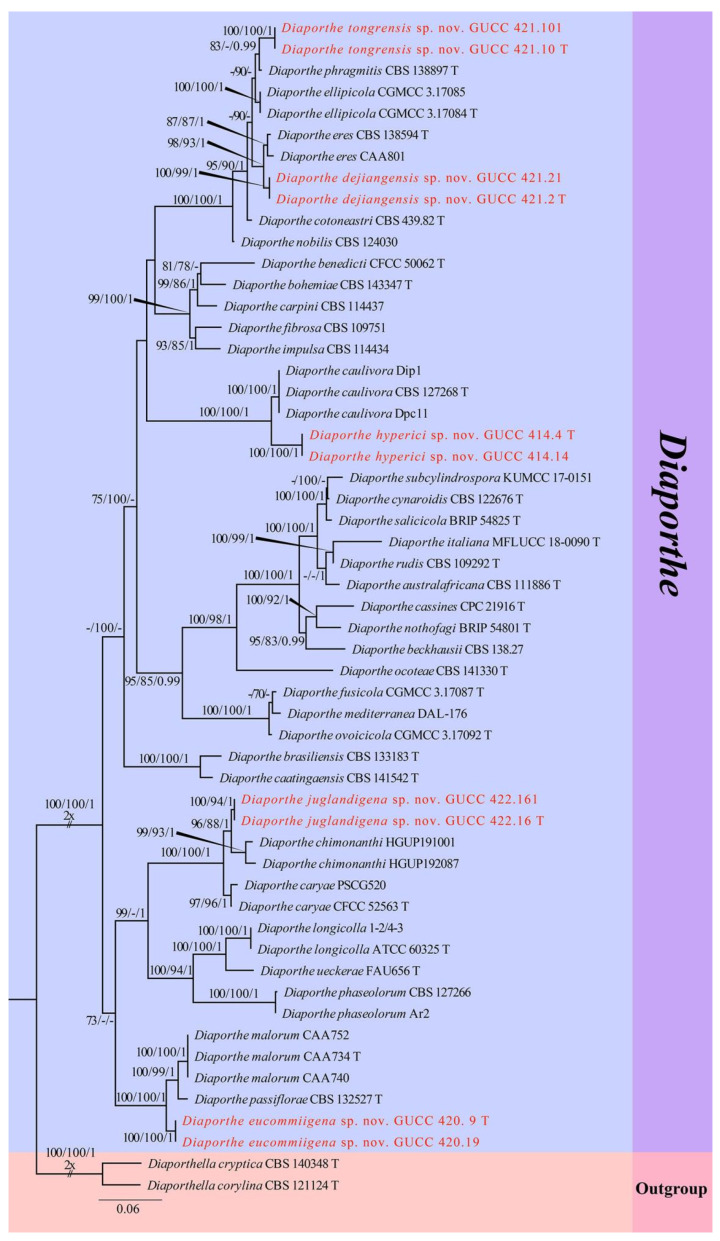
Phylogram generated from RAxML analysis of a concatenated ITS-*tef1*-*tub2* sequence dataset to represent the phylogenetic relationships of taxa in *Diaporthe*. The tree was rooted with *Diaporthella corylina* (CBS 121124, ex-type strain) and *Di. cryptica* (CBS 140348, ex-neotype strain). Bootstrap support values for ML and MP equal to or greater than 70% and Bayesian posterior probabilities equal to or higher than 0.95 PP are indicated above the nodes as ML/MP/PP. Support values lower than 70% ML/MP and 0.95 PP are indicated by a hyphen (-). The newly generated sequences are indicated in red.

**Figure 4 jof-08-01301-f004:**
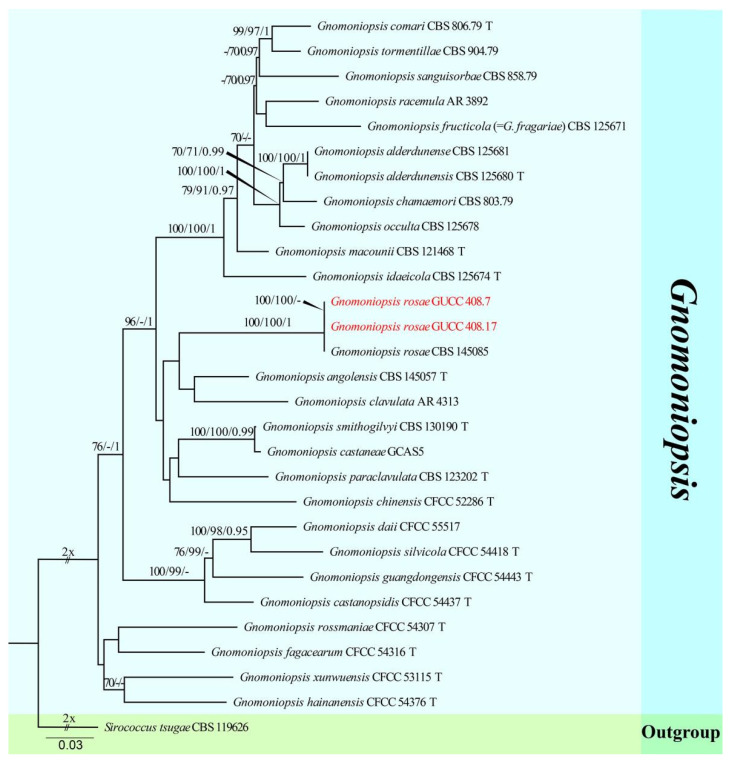
Phylogram generated from RAxML analysis of a concatenated ITS-LSU-*rpb2*-*tef1*-*tub2* sequence dataset to represent the phylogenetic relationships of taxa in *Gnomoniopsis*. The tree was rooted with *Sirococcus tsugae* (CBS 119626). Bootstrap support values for ML and MP equal to or greater than 70% and the Bayesian posterior probabilities equal to or higher than 0.95 PP are indicated above the nodes as ML/MP/PP. Support values lower than 70% ML/MP and 0.95 PP are indicated by a hyphen (-). The newly generated sequences are indicated in red.

**Figure 5 jof-08-01301-f005:**
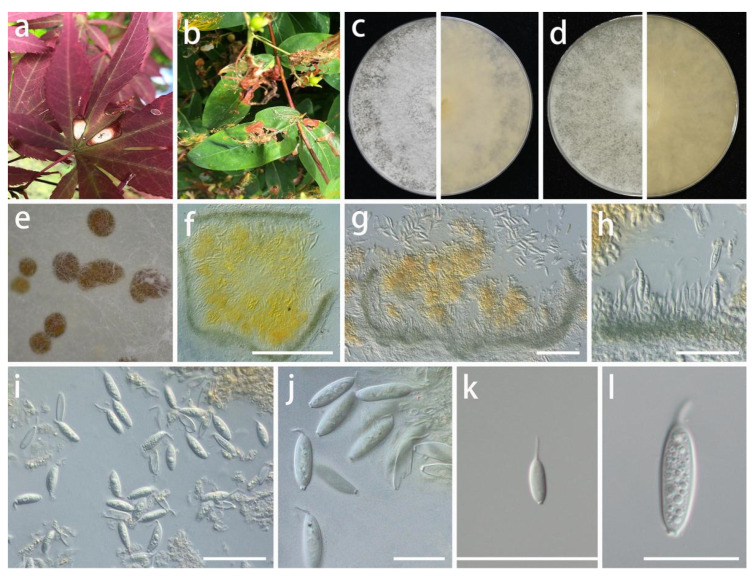
*Pseudomastigosporella guizhouensis* (GUCC 406.6). Hosts: (**a**) *Hypericum patulum*; (**b**) *Acer palmatum*. (**c**) Colony on PDA after 2 wk at 25 °C (left: above, right: reverse). (**d**) Colony on OA after 2 wk at 25 °C (left: above, right: reverse). (**e**) Mass of conidia. (**f**) Conidioma. (**g**,**h**) Conidiomata and conidiogenous cells. (**i**–**l**) Conidia. Scale bars: (**f**–**i**) = 50 µm; (**j**–**l**) = 10 µm.

**Figure 6 jof-08-01301-f006:**
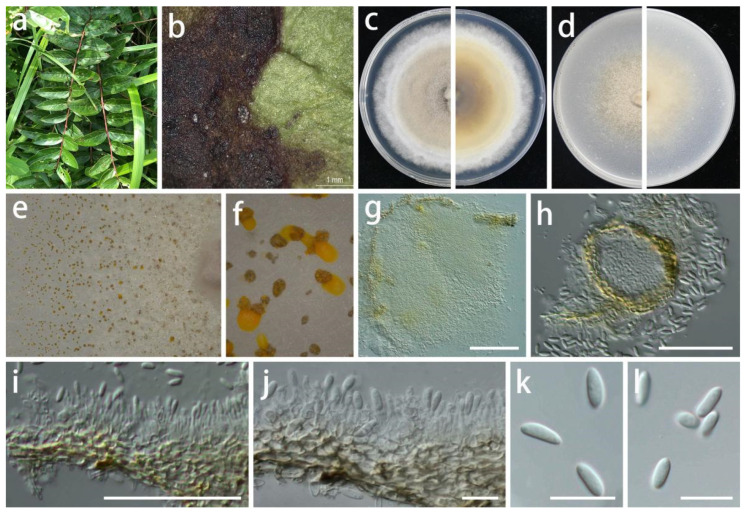
*Chrysofolia coriariae* (GUCC 416.4). (**a**,**b**) Host: *Coriaria nepalensis*. (**c**) Colony on PDA after 2 wk at 25 °C (left: above, right: reverse). (**d**) Colony on OA after 2 wk at 25 °C (left: above, right: reverse). (**e**,**f**) Conidial masses. (**g**,**h**) Conidiomata. (**i**,**j**) Conidiogenous cells. (**k**,**l**) Conidia. Scale bars: (**g**–**i**) = 50 µm; (**j**–**l**) = 10 µm.

**Figure 7 jof-08-01301-f007:**
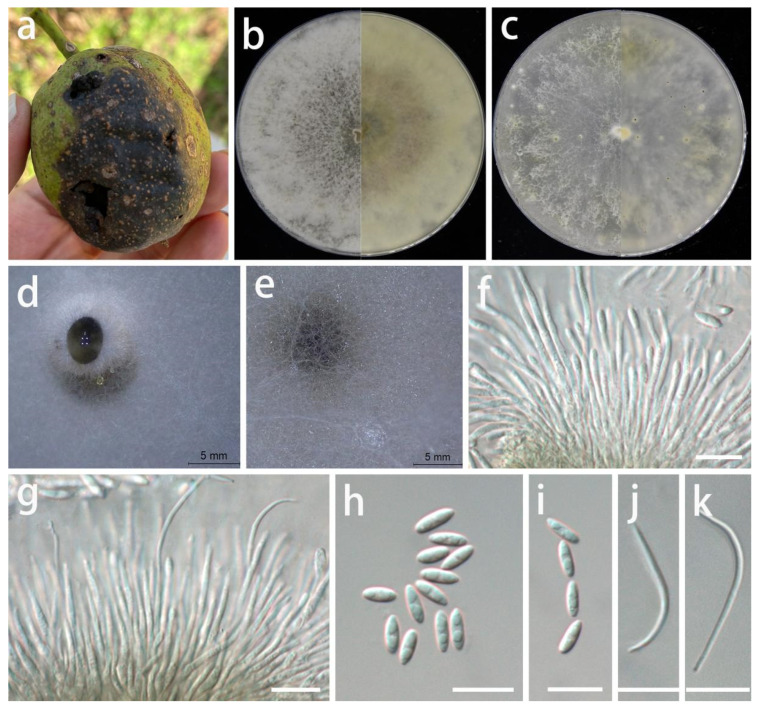
*Diaporthe juglandigena* (GUCC 422.16). (**a**) Host: *Juglans regia*. (**b**) Colony on PDA after 2 wk (left: above, right: reverse). (**c**) Colony on OA after 2 wk (left: above, right: reverse). (**d**,**e**) Mass of conidia. (**f**,**g**) Conidiogenous cells. (**h**,**i**) Alpha conidia. (**j**,**k**) Beta conidia. Scale bars: (**f**–**k**) = 10 µm.

**Figure 8 jof-08-01301-f008:**
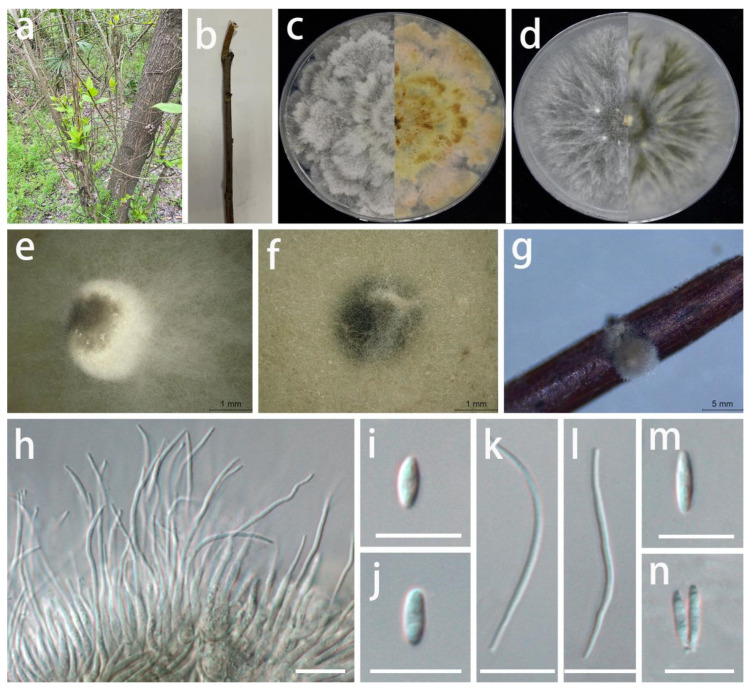
*Diaporthe eucommiigena* (GUCC 420.9). (**a**,**b**) Host: *Eucommia ulmoides*. (**c**) Colony on PDA after 2 wk at 25 °C (left: above, right: reverse). (**d**) Colony on OA after 2 wk at 25 °C (left: above, right: reverse). (**e**–**g**) Mass of conidia. (**h**) Conidiogenous cells. (**i**,**j**) Alpha conidia. (**k**,**l**) Beta conidia. (**m**,**n**) Gamma conidia. Scale bars: (**h**–**n**) = 10 µm.

**Figure 9 jof-08-01301-f009:**
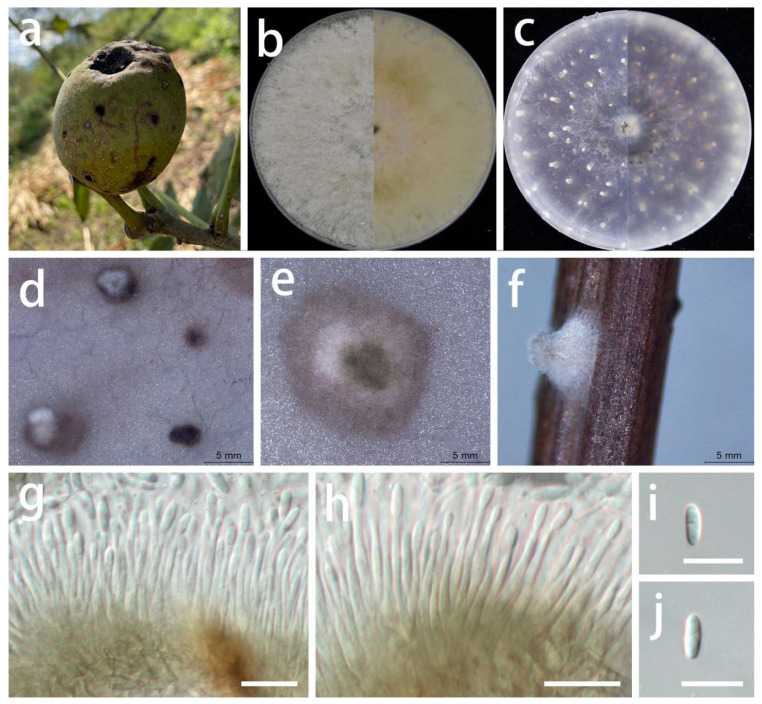
*Diaporthe dejiangensis* (GUCC 421.2). (**a**) Host: *Juglans regia*. (**b**) Colony on PDA after 2 wk at 25 °C (left: above, right: reverse). (**c**) Colony on OA after 2 wk at 25 °C (left: above, right: reverse). (**d**–**f**) Mass of conidia. (**g**,**h**) Conidiogenous cells. (**i**,**j**) Alpha conidia. Scale bars: (**g**–**j**) = 10 µm.

**Figure 10 jof-08-01301-f010:**
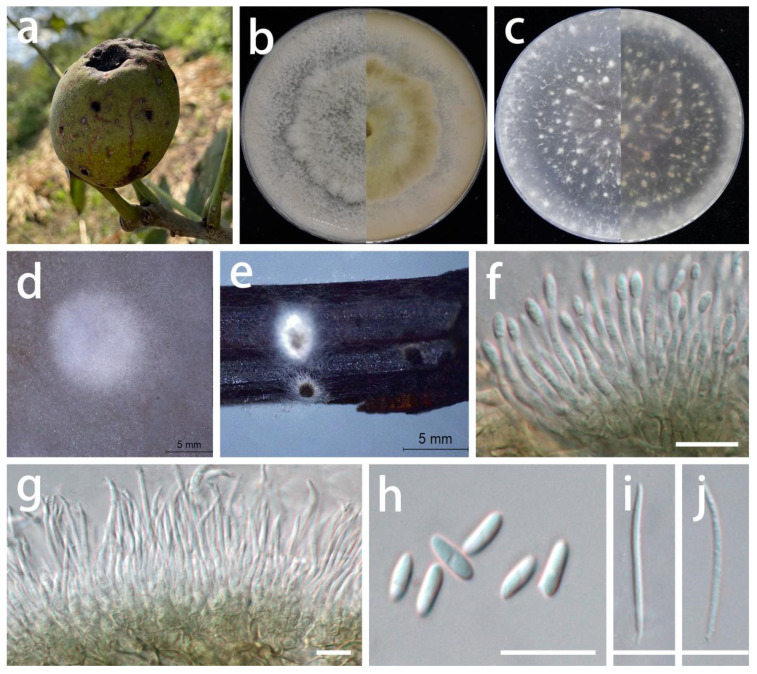
*Diaporthe tongrensis* (GUCC 421.10). (**a**) Host: *Juglans regia*. (**b**) Colony on PDA after 2 wk at 25 °C (left: above, right: reverse). (**c**) Colony on OA after 2 wk at 25 °C (left: above, right: reverse). (**d**,**e**) Mass of conidia. (**f**,**g**) Conidiogenous cells. h Alpha conidia. (**i**,**j**) Beta conidia. Scale bars: (**f**–**j**) = 10 µm.

**Figure 11 jof-08-01301-f011:**
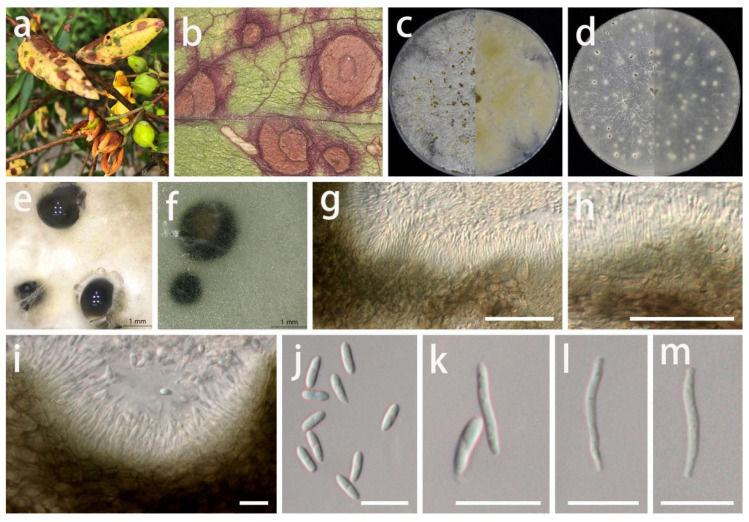
*Diaporthe hyperici* (GUCC 414.4). (**a**,**b**) Host: *Hypericum patulum*. (**c**) Colony on PDA after 2 wk at 25 °C (left: above, right: reverse). (**d**) Colony on OA after 2 wk at 25 °C (left: above, right: reverse). (**e**,**f**) Mass of conidia. (**g**–**i**) Conidiogenous cells. (**j**) Alpha conidia. (**k**) Alpha and beta conidia. (**l**,**m**) Beta conidia. Scale bars: (**g**,**h**) = 50 µm; (**i**–**m**) = 10 µm.

**Figure 12 jof-08-01301-f012:**
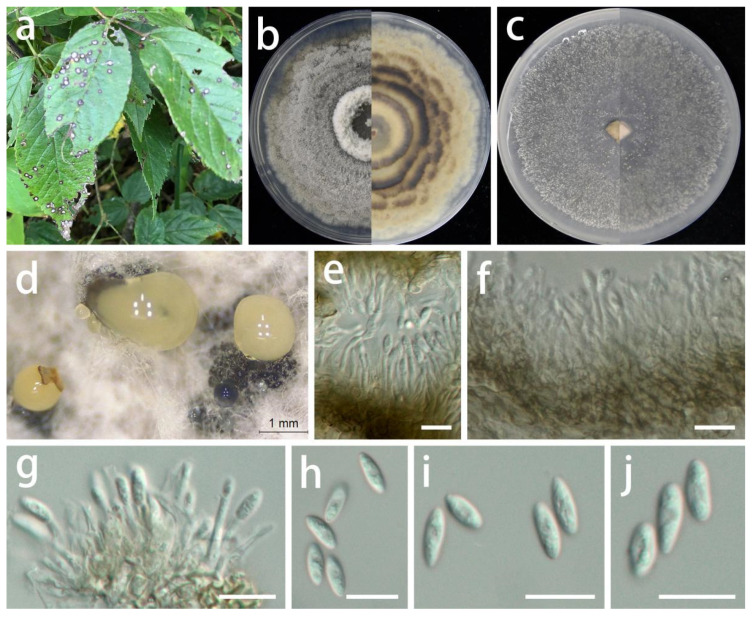
*Gnomoniopsis rosae* (GUCC 408.7). (**a**) Host: *Rose* sp. (**b**) Colonies on PDA after 2 wk at 25 °C (left: above, right: reverse). (**c**) Colony on OA after 2 wk at 25 °C (left: above, right: reverse). (**d**) Mass of conidia. (**e**) Conidiomata. (**f**,**g**) Conidiogenous cells. (**h**–**j**) Conidia. Scale bars: (**e**–**j**) = 10 µm.

**Figure 13 jof-08-01301-f013:**
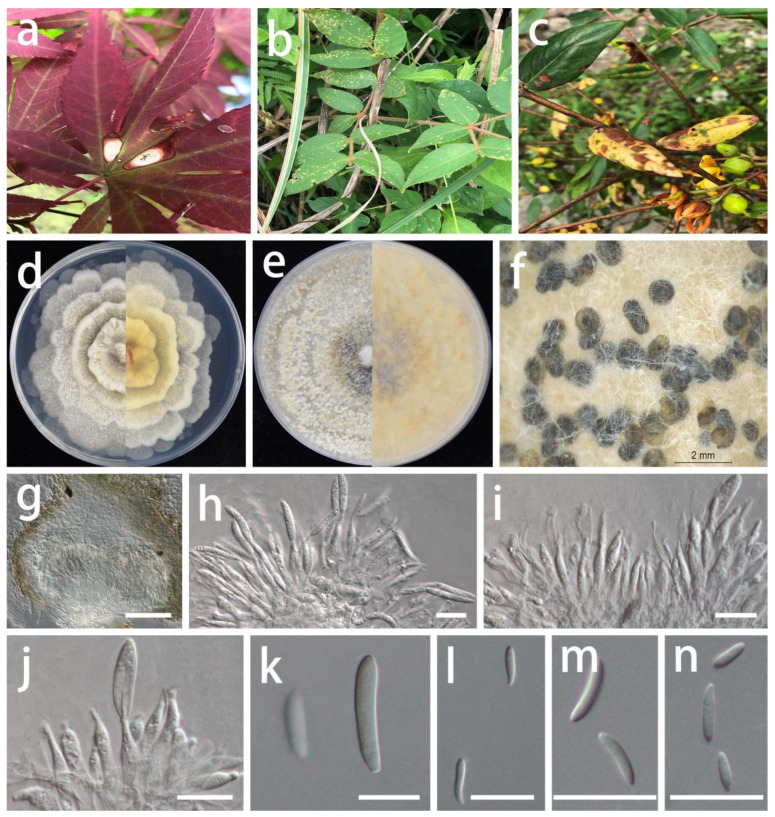
*Coniella quercicola* (GUCC 412.3). Hosts. (**a**) *Acer palmatum*; (**b**) *Aralia chinensis*; (**c**) *Hypericum patulum*. (**d**) Colony on PDA after 2 wk 25 °C (left: above, right: reverse). (**e**) Colony on OA after 2 wk 25 °C (left: above, right: reverse). (**f**) Mass of conidia. (**g**) Conidioma. (**h**–**j**) Conidiogenous cells. (**k**–**n**) Conidia. Scale bars: (**g**) = 50 µm; (**h**–**n**) = 10 µm.

**Figure 14 jof-08-01301-f014:**
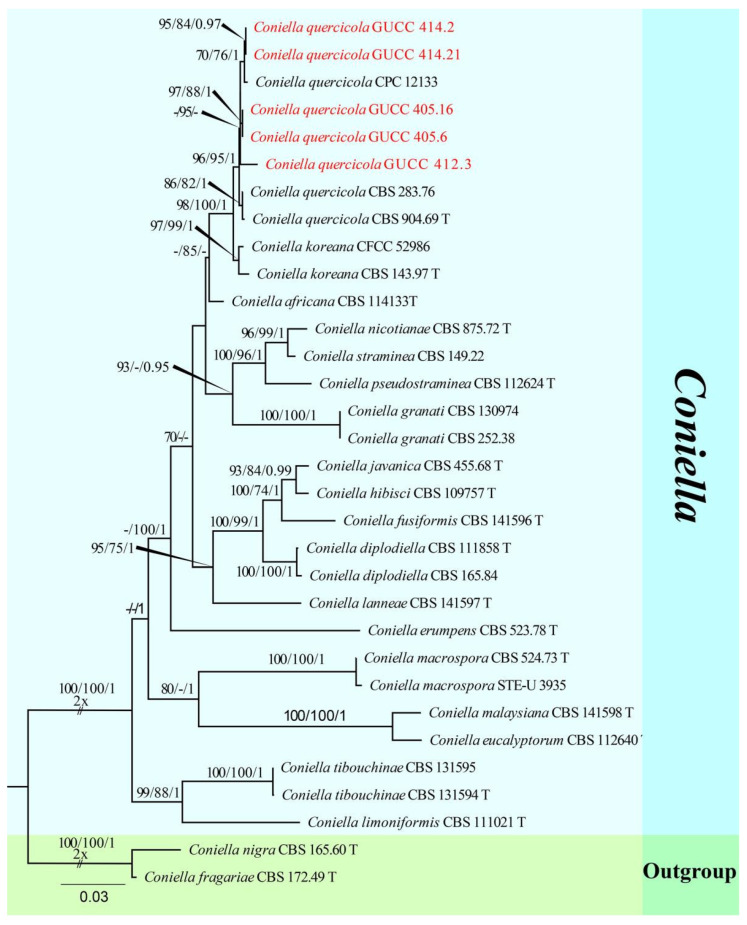
Phylogram generated from RAxML analysis of a concatenated ITS-LSU-*tef1* sequence dataset to represent the phylogenetic relationships of taxa in *Coniella*. The tree was rooted with *C. fragariae* (CBS 172.49, ex-type strain) and *C. nigra* (CBS 165.60, ex-type strain). Bootstrap support values for ML and MP equal to or greater than 70% and the Bayesian posterior probabilities equal to or higher than 0.95 PP are indicated above the nodes as ML/MP/PP. Support values lower than 70% ML/MP and 0.95 PP are indicated by a hyphen (-). The newly generated sequences are indicated in red.

**Table 1 jof-08-01301-t001:** Primers and PCR procedures used in this study.

Locus	Primers	Primer Sequence (5′–3′)	PCR Thermal Cycle Protocols	References
ITS	ITS5	GGAAGTAAAAGTCGTAACAAGG	94 °C–2 min; 94 °C–1 min; 58 °C–1 min; 72 °C–90 s; repeat 2–4 for 35 cycles; 72 °C–10 min; 4 °C on hold	White et al. [23]
	ITS4	TCCTCCGCTTATTGATATGC		
LSU	LR0R	ACCCGCTGAACTTAAGC		Rehner and Samuels [24]; Vilgalys and Hester [25]
	LR5	TCCTGAGGGAAACTTCG		
*rpb2*	fRPB2-5F	GAYGAYMGWGATCAYTTYGG	94 °C–2 min; 95 °C–45 s; 57 °C–50 s; 72 °C–90 s; repeat 2–4 for 35 cycles; 72 °C–10 min; 4 °C on hold	Liu et al. [26]
	fRPB2-7cR	CCCATRGCTTGYTTRCCCAT		
*tef1*	EF1-728F	CATCGAGAAGTTCGAGAAGG	94 °C–2 min; 95 °C–30 s; 58 °C–50 s; 72 °C–1 min; repeat 2–4 for 35 cycles; 72 °C–10 min; 4 °C on hold	Carbone and Kohn [27]
	EF1-986R	TACTTGAAGGAACCCTTACC		
*tub2*	Bt2a	GGTAACCAAATCGGTGCTGCTTTC	95 °C–5 min; 95 °C–30 s; 60 °C–30 s; 72 °C–30 s; repeat 2–4 for 34 cycles; 72 °C–10 min; 4 °C on hold	Glass and Donaldson [28]
	Bt2b	ACCCTCAGTGTAGTGACCCTTGGC		

**Table 2 jof-08-01301-t002:** Strains and their GenBank accession numbers used in the molecular phylogenetic analyses of *Cryphonectriaceae* and *Foliocryphiaceae*. Newly generated sequences are in bold. (^T^) = ex–type strain.

Species	Strain Number	GenBank Accession Number				
		ITS	LSU	*rpb2*	*tef1*	*tub2*
*Amphilogia gyrosa*	CBS 112922^T^	AF452111	AY194107	MN271782	MN271818	AF525714
*Aurantioporthe corni*	CBS 245.90	MN172403	MN172371	MN271784	MN271822	–
*Aurantiosacculus acutatus*	CBS 132181^T^	JQ685514	JQ685520	–	MN271823	–
*Aurantiosacculus castaneae*	CFCC 52456^T^	MH514025	MH514015	MN271786	–	MH539688
*Aurantiosacculus eucalyptorum*	CPC 13229^T^	JQ685515	JQ685521	MN271785	MN271824	–
*Aurapex penicillata*	CMW 10030^T^	AY214311	AY194103	MN271787	–	AY214239
*Aurifilum marmelostoma*	CBS 124928^T^	MH863426	MH874934	MN271788	MN271827	MN987000
*Aurifilum terminali*	CSF10762	MN199838	MN258786	–	MN258781	MN258771
*Celoporthe dispersa*	CBS 118782^T^	DQ267130	HQ730853	–	HQ730840	AY214280
*Celoporthe eucalypti*	CMW 26908^T^	HQ730837	HQ730863	MN271790	HQ730850	MN263386
*Chrysofolia barringtoniae*	TBRC 5647	KU948046	KU948045	–	–	–
*Chrysofolia colombiana*	CPC 24986^T^	KR476738	KR476771	–	MN271829	–
** *Chrysofolia coriariae* ** **sp. nov.**	**GUCC 416.4^T^**	**OP581211**	**OP581237**	**–**	**OP688516**	**OP688542**
** *Chrysofolia coriariae* ** **sp. nov.**	**GUCC 416.14**	**OP581212**	**OP581238**	**–**	**OP688517**	**OP688543**
*Chrysomorbus lagerstroemiae*	CBS 142594^T^	KY929338	KY929328	–	MN271830	KY929348
*Chrysoporthe hodgesiana*	CBS 115854^T^	AY692322	MN172380	MN271793	MN271836	–
*Chrysoporthe syzygiicola*	CBS 124488^T^	FJ655005	MN172383	–	MN271839	FJ805236
*Corticimorbus sinomyrti*	CBS 140205^T^	KT167169	KT167179	MN271794	MN271841	KT167183
*Cryphonectria macrospora*	CBS 109764	EU199182	AF408340	EU220029	KC465405	AH014594
*Cryphonectria parasitica*	ATCC 38755	MH843497	MH514021	DQ862017	MN271848	MW086477
*Cryptometrion aestuescens*	CBS 124014^T^	MH863335	MH874864	MN271798	MN271851	GQ369456
*Diversimorbus metrosideri*	CBS 132866^T^	JQ862871	JQ862828	–	MN271857	JQ862953
*Dwiroopa lythri*	CBS 109755^T^	MN172410	MN172389	MN271801	MN271859	–
*Dwiroopa punicae*	CBS 143163^T^	MK510676	MK510686	MK510692	MH020056	MK510714
*Endothia chinensis*	CFCC 52144^T^	MH514027	MH514017	–	MN271860	MH539690
*Foliocryphia eucalypti*	CBS 124779^T^	GQ303276	GQ303307	MN271802	MN271861	JQ706128
*Foliocryphia eucalyptorum*	CBS 142536^T^	KY979772	KY979827	MN271803	MN271862	KY979936
*Holocryphia capensis*	CBS 132870^T^	JQ862854	JQ862811	–	MN271883	JQ862948
*Holocryphia eucalypti*	CBS 115842^T^	MN172411	MN172391	MN271804	MN271882	JQ862772
*Immersiporthe knoxdaviesiana*	CBS 132862^T^	JQ862765	JQ862755	MN271805	MN271886	JQ862785
*Latruncellus aurorae*	CBS 125526^T^	GU726947	HQ730872	–	MN271888	GU726959
*Luteocirrhus shearii*	CBS 130776^T^	KC197021	KC197019	MN271807	MN271890	KC197006
*Microthia havanensis*	CBS 115855	DQ368735	MN172393	MN271811	–	–
*Myrtonectria myrtacearum*	CMW 46433^T^	MG585736	MG585750	–	–	MG585734
*Neocryphonectria carpini*	CFCC 53027^T^	MN172413	MN172396	–	–	–
*Neocryphonectria chinensis*	CFCC 53025^T^	MN172414	MN172397	MN271812	MN271893	–
*Neocryphonectria chinensis*	CFCC 53029	MN172415	MN172398	MN271813	MN271894	–
***Pseudomastigosporella guizhouensis* sp. nov.**	**GUCC 406.6^T^**	**OP581233**	**OP581246**	**OP688514**	**OP688538**	**OP688563**
***Pseudomastigosporella guizhouensis* sp. nov.**	**GUCC 405.3**	**OP581234**	**OP581247**	**OP688515**	**OP688539**	**OP688564**
***Pseudomastigosporella guizhouensis* sp. nov.**	**GUCC 405.4**	**OP581235**	**OP581248**	**–**	**OP688540**	**OP688565**
***Pseudomastigosporella guizhouensis* sp. nov.**	**GUCC 405.8**	**OP581236**	**OP581249**	**–**	**OP688541**	**OP688566**
*Rostraureum tropicale*	CBS 115725^T^	AY167435	MN172399	MN271814	MN271895	AY167431
*Ursicollum fallax*	CBS 118663^T^	DQ368755	EF392860	MN271816	MN271897	AH015658

**Table 3 jof-08-01301-t003:** Strains and their GenBank accession numbers used in the molecular phylogenetic analyses of *Diaporthe*. Newly generated sequences are in bold. (^T^) = ex–type strain.

Species	Strain Number	GenBank Accession Number		
		ITS	*tef1*	*tub2*
*Diaporthe australafricana*	CBS 111886^T^	KC343038	KC343764	KC344006
*D. beckhausii*	CBS 138.27	KC343041	KC343767	KC344009
*D. benedicti*	CFCC 50062^T^	KP208847	KP208853	KP208855
*D. bohemiae*	CBS 143347^T^	MG281015	MG281536	MG281188
*D. brasiliensis*	CBS 133183^T^	KC343042	KC343768	KC344010
*D. caatingaensis*	CBS 141542^T^	KY085927	KY115603	KY115600
*D. carpini*	CBS 114437	KC343044	KC343770	KC344012
*D. caryae*	CFCC 52563^T^	MH121498	MH121540	MH121580
*D. caryae*	PSCG520	MK626952	MK654895	MK691315
*D. cassines*	CPC 21916^T^	KF777155	KF777244	–
*D. caulivora*	CBS 127268^T^	KC343045	KC343771	KC344013
*D. caulivora*	Dip1	HM347703	HM347687	–
*D. caulivora*	Dpc11	HM347704	HM347688	–
*D. chimonanthi*	HGUP191001	MZ724752	–	MZ724033
*D. chimonanthi*	HGUP192087	MZ724753	–	MZ724034
*D. cotoneastri*	CBS 439.82^T^	MH861511	GQ250341	JX275437
*D. cynaroidis*	CBS 122676^T^	KC343058	KC343784	KC344026
***D. dejiangensis* sp. nov.**	**GUCC 421.2^T^**	**OP581221**	**OP688526**	**OP688551**
***D. dejiangensis* sp. nov.**	**GUCC 421.21**	**OP581222**	**OP688527**	**OP688552**
*D. ellipicola*	CGMCC 3.17084^T^	KF576270	KF576245	KF576294
*D. ellipicola*	CGMCC 3.17085	KF576271	KF576246	KF576295
*D. eres*	CBS 138594^T^	KJ210529	KJ210550	KJ420799
*D. eres*	CAA801	KY435644	KY435631	KY435672
** *D. eucommiigena* ** **sp. nov.**	**GUCC 420.9^T^**	**OP581223**	**OP688528**	**OP688553**
** *D. eucommiigena* ** **sp. nov.**	**GUCC 420.19**	**OP581224**	**OP688529**	**OP688554**
*D. fibrosa*	CBS 109751	KC343099	KC343825	KC344067
*D. fusicola*	CGMCC 3.17087^T^	KF576281	KF576256	KF576305
** *D. hyperici* ** **sp. nov.**	**GUCC 414.4^T^**	**OP581227**	**OP688532**	**OP688557**
** *D. hyperici* ** **sp. nov.**	**GUCC 414.14**	**OP581228**	**OP688533**	**OP688558**
*D. impulsa*	CBS 114434	KC343121	KC343847	KC344089
*D. italiana*	MFLUCC 18-0090^T^	MH846237	MH853686	MH853688
***D. juglandigena* sp. nov.**	**GUCC 422.16^T^**	**OP581229**	**OP688534**	**OP688559**
***D. juglandigena* sp. nov.**	**GUCC 422.161**	**OP581230**	**OP688535**	**OP688560**
*D. longicolla*	ATCC 60325^T^	KJ590728	KJ590767	KJ610883
*D. longicolla*	1-2/4-3	HM347711	HM347682	–
*D. malorum*	CAA734^T^	KY435638	KY435627	KY435668
*D. malorum*	CAA740	KY435642	KY435629	KY435670
*D. malorum*	CAA752	KY435643	KY435630	KY435671
*D. mediterranea*	DAL-176	MT007496	MT006996	MT006693
*D. nobilis*	CBS 124030	KC343149	KC343875	KC344117
*D. nothofagi*	BRIP 54801^T^	JX862530	JX862536	KF170922
*D. ocoteae*	CBS 141330^T^	KX228293	–	KX228388
*D. ovoicicola*	CGMCC 3.17092^T^	KF576264	KF576239	KF576288
*D. passiflorae*	CBS 132527^T^	JX069860	KY435633	KY435674
*D. phaseolorum*	Ar2	HM347705	HM347679	–
*D. phaseolorum*	CBS 127266	HM347707	HM347672	HQ333513
*D. phragmitis*	CBS 138897^T^	KP004445	–	KP004507
*D. rudis*	CBS 109292^T^	KC343234	KC343960	KC344202
*D. salicicola*	BRIP 54825^T^	JX862531	JX862537	KF170923
*D. subcylindrospora*	KUMCC 17-0151	MG746629	MG746630	MG746631
** *D. tongrensis* ** **sp. nov.**	**GUCC 421.10^T^**	**OP581225**	**OP688530**	**OP688555**
** *D. tongrensis* ** **sp. nov.**	**GUCC 421.101**	**OP581226**	**OP688531**	**OP688556**
*D. ueckerae*	FAU656^T^	KJ590726	KJ590747	KJ610881
*Diaporthella corylina*	CBS 121124^T^	KC343004	KC343730	KC343972
*Diaporthella cryptica*	CBS 140348^T^	MN172409	MN271854	–

**Table 4 jof-08-01301-t004:** Strains and their GenBank accession numbers used in the molecular phylogenetic analyses of *Gnomoniopsis*. Newly generated sequences are in bold. (^T^) = ex–type strain.

Species	Strain Number	GenBank Accession Number				
		ITS	LSU	*rpb2*	*tef1*	*tub2*
*Gnomoniopsis alderdunense*	CBS 125681	GU320827	MH875098	–	GU320802	GU320789
*G. alderdunensis*	CBS 125680^T^	MH863625	MH875097	–	GU320801	GU320787
*G. angolensis*	CBS 145057^T^	MK047428	MK047479	MK047539	–	–
*G. castaneae*	GCAS5	MH107830	MZ682110	–	MH213486	MH213481
*G. castanopsidis*	CFCC 54437^T^	MZ902909	–	–	MZ936385	
*G. chamaemori*	CBS 803.79	EU254808	EU255107	–	GU320809	EU219155
*G. chinensis*	CFCC 52286^T^	MG866032	–	–	MH545370	MH545366
*G. clavulata*	AR 4313	EU254818	–	EU219251	EU221934	EU219211
*G. comari*	CBS 806.79^T^	EU254821	EU255114	EU219286	GU320810	EU219156
*G. daii*	CFCC 55517	MZ902911	–	–	MZ936387	MZ936403
*G. fagacearum*	CFCC 54316^T^	MZ902916	–	–	MZ936392	MZ936408
*G. fructicola* (=*G. fragariae*)	CBS 125671	MH863616	MH875088	–	GU320793	GU320776
*G. guangdongensis*	CFCC 54443^T^	MZ902918	–	–	MZ936394	MZ936410
*G. hainanensis*	CFCC 54376^T^	MZ902921	–	–	MZ936397	MZ936413
*G. idaeicola*	CBS 125674^T^	MH863619	MH875091	–	GU320796	GU320780
*G. macounii*	CBS 121468^T^	MH863110	MH874666	–	GU320804	–
*G. occulta*	CBS 125678	MH863623	MH875095	–	GU320800	GU320786
*G. paraclavulata*	CBS 123202^T^	GU320830	–	–	GU320815	GU320775
*G. racemula*	AR 3892	EU254841	EU255122	EU219241	EU221889	EU219125
*G. rosae*	CBS 145085	MK047451	MK047501	MK047547	–	–
** *G. rosae* **	**GUCC 408.7**	**OP581231**	**OP581244**	**OP688512**	**OP688536**	**OP688561**
** *G. rosae* **	**GUCC 408.17**	**OP581232**	**OP581245**	**OP688513**	**OP688537**	**OP688562**
*G. rossmaniae*	CFCC 54307^T^	MZ902923	–	–	MZ936399	MZ936415
*G. sanguisorbae*	CBS 858.79	GU320818	KY496735	–	GU320805	GU320790
*G. silvicola*	CFCC 54418^T^	MZ902926	–	–	MZ936402	MZ936418
*G. smithogilvyi*	CBS 130190^T^	MH865607	MH877031	JQ910648	JQ910645	JQ910639
*G. tormentillae*	CBS 904.79	EU254856	EU255133	–	GU320795	EU219165
*G. xunwuensis*	CFCC 53115T	MK432667	MK429910	–	MK578141	MK578067
*Sirococcus tsugae*	CBS 119626	EF512472	–	–	EF512534	EU219140

**Table 5 jof-08-01301-t005:** Strains and their GenBank accession numbers used in the molecular phylogenetic analyses of *Coniella*. Newly generated sequences are in bold. (^T^) = ex–type strain.

Species	Strain Number	GenBank Accession Number		
		ITS	LSU	*tef1*
*Coniella africana*	CBS 114133^T^	AY339344	AY339293	KX833600
*C. diplodiella*	CBS 111858^T^	MH862886	KX833335	KX833603
*C. diplodiella*	CBS 165.84	KX833529	KX833354	KX833622
*C. erumpens*	CBS 523.78^T^	KX833535	KX833361	KX833630
*C. eucalyptorum*	CBS 112640^T^	AY339338	AY339290	KX833637
*C. fusiformis*	CBS 141596^T^	KX833576	KX833397	KX833674
*C. granati*	CBS 130974	JN815312	KX833398	KX833675
*C. granati*	CBS 252.38	KX833581	AY339291	KX833681
*C. hibisci*	CBS 109757^T^	KX833589	AF408337	KX833689
*C. javanica*	CBS 455.68^T^	KX833583	KX833403	KX833683
*C. koreana*	CBS 143.97^T^	KX833584	AF408378	KX833684
*C. koreana*	CFCC 52986	MK432612	MK429882	MK578112
*C. lanneae*	CBS 141597^T^	KX833585	KX833404	KX833685
*C. limoniformis*	CBS 111021^T^	KX833586	KX833405	KX833686
*C. macrospora*	CBS 524.73^T^	KX833587	AY339292	KX833687
*C. macrospora*	STE-U 3935	AY339343	–	AY339363
*C. malaysiana*	CBS 141598^T^	KX833588	KX833406	KX833688
*C. nicotianae*	CBS 875.72^T^	KX833590	KX833407	KX833690
*C. pseudostraminea*	CBS 112624^T^	KX833593	KX833412	KX833696
*C. quercicola*	CBS 904.69^T^	KX833595	KX833414	KX833698
*C. quercicola*	CBS 283.76	KX833594	KX833413	KX833697
*C. quercicola*	CPC 12133	KX833596	–	KX833699
** *C. quercicola* **	**GUCC 414.2**	**OP581213**	**OP581239**	**OP688518**
** *C. quercicola* **	**GUCC 414.21**	**OP581214**	**OP581240**	**OP688519**
** *C. quercicola* **	**GUCC 412.3**	**OP581215**	**OP581241**	**OP688520**
** *C. quercicola* **	**GUCC 405.6**	**OP581216**	**OP581242**	**OP688521**
** *C. quercicola* **	**GUCC 405.16**	**OP581217**	**OP581243**	**OP688522**
*C. straminea*	CBS 149.22	AY339348	AY339296	KX833704
*C. tibouchinae*	CBS 131594^T^	JQ281774	KX833418	JQ281778
*C. tibouchinae*	CBS 131595	JQ281775	KX833419	JQ281779
*C. fragariae*	CBS 172.49^T^	AY339317	AY339282	KX833663
*C. nigra*	CBS 165.60^T^	AY339319	KX833408	KX833691

**Table 6 jof-08-01301-t006:** Parameters of maximum parsimony and Bayesian methods in this study.

Datasets	Maximum Parsimony (MP)					
	TL	PT	CI	RI	RC	HI
*Foliocryphiaceae*	4515	9	0.5132	0.6746	0.3462	0.4868
*Diaporthe*	2261	9	0.5657	0.8362	0.4730	0.4343
*Gnomoniopsis*	2423	3	0.5712	0.5930	0.3387	0.4288
*Coniella*	1292	8	0.5735	0.7112	0.4079	0.4265
	**Bayesian**					
	**Model**					**ASDSF**
	**ITS**	**LSU**	** *rpb2* **	** *tef1* **	** *tub2* **	
*Foliocryphiaceae*	GTR+I+G			HKY+I+G	HKY+G	0.009523
*Diaporthe*	SYM+I+G	n/a	n/a	HKY+I+G		0.009906
*Gnomoniopsis*	GTR+I+G	GTR+I	GTR+G	HKY+I+G	GTR+I+G	0.009908
*Coniella*	SYM+I+G	GTR+I+G	n/a	GTR+I+G	n/a	0.009408

TL: Tree length; PT: Parsimonious tree; CI: Consistency indices; RI: Retention indices; RC: Rescaled consistency indices; HI: Homoplasy index; Model: the best nucleotide substitution model used for the different partitions; ASDSF: average standard deviation of split frequencies.

**Table 7 jof-08-01301-t007:** DNA base differences between our strains and related taxa in the five-locus regions. Asterisks (*) denote our material. (^T^) = ex–type strain.

Species	Strain Number	Gene Region and Alignment Positions				
		ITS (1–769 Characters)	LSU (770–1617 Characters)	*rpb2* (1618–2442 Characters)	*tef1* (2443–2952 Characters)	*tub2* (2953–3596 Characters)
		**(with gap: 151 characters)**	**(with gap: 41 characters)**	**(no data)**	**(no data)**	**(no data)**
*Chrysofolia barringtoniae*	TBRC 5647	–	–	n/a	n/a	n/a
*Chrysofolia coriariae* sp. nov. *	GUCC 416.4^T^	31	4	n/a	n/a	n/a
*Chrysofolia coriariae* sp. nov. *	GUCC 416.14	31	4	n/a	n/a	n/a
		**(with gap: 158 characters)**	**(with gap: 41 characters)**	**(no data)**	**(with gap: 156 characters)**	**(no data)**
*Chrysofolia colombiana*	CPC 24986	–	–	n/a	–	n/a
*Chrysofolia coriariae* sp. nov. *	GUCC 416.4^T^	7	2	n/a	149	n/a
*Chrysofolia coriariae* sp. nov. *	GUCC 416.14	7	2	n/a	149	n/a
		**(with gap: 152 characters)**	**(with gap: 34 characters)**	**(no data)**	**(no data)**	**(no data)**
*Neocryphonectria carpini*	CFCC 53027^T^	–	–	n/a	n/a	n/a
*Pseudomastigosporella guizhouensis* sp. nov. *	GUCC 406.6^T^	116	22	n/a	n/a	n/a
*Pseudomastigosporella guizhouensis* sp. nov. *	GUCC 405.3	116	21	n/a	n/a	n/a
*Pseudomastigosporella guizhouensis* sp. nov. *	GUCC 405.4	116	21	n/a	n/a	n/a
*Pseudomastigosporella guizhouensis* sp. nov. *	GUCC 405.8	116	21	n/a	n/a	n/a
		**(with gap: 139 characters)**	**(with gap: 34 characters)**	**(with gap: 53 characters)**	**(with gap: 177 characters)**	**(no data)**
*Neocryphonectria chinensis*	CFCC 53025^T^	–	–	–	–	n/a
*Pseudomastigosporella guizhouensis* sp. nov. *	GUCC 406.6^T^	118	21	75	130	n/a
*Pseudomastigosporella guizhouensis* sp. nov. *	GUCC 405.3	118	20	76	131	n/a
*Pseudomastigosporella guizhouensis* sp. nov. *	GUCC 405.4	118	20	n/a	129	n/a
*Pseudomastigosporella guizhouensis* sp. nov. *	GUCC 405.8	118	20	n/a	132	n/a
		**(with gap: 333 characters)**	**(with gap: 34 characters)**	**(with gap: 200 characters)**	**(with gap: 177 characters)**	**(no data)**
*Neocryphonectria chinensis*	CFCC 53029	–	–	–	–	n/a
*Pseudomastigosporella guizhouensis* sp. nov. *	GUCC 406.6^T^	32	21	66	130	n/a
*Pseudomastigosporella guizhouensis* sp. nov. *	GUCC 405.3	32	20	67	131	n/a
*Pseudomastigosporella guizhouensis* sp. nov. *	GUCC 405.4	32	20	n/a	129	n/a
*Pseudomastigosporella guizhouensis* sp. nov. *	GUCC 405.8	32	20	n/a	132	n/a
		**ITS** **(1–640 characters)**	**(no data)**	**(no data)**	** *tef1* ** **(641–1116 characters)**	** *tub2* ** **(1117–1770 characters)**
		**(with gap: 146 characters)**	**(no data)**	**(no data)**	**(with gap: 163 characters)**	**(with gap: 140 characters)**
*Diaporthe juglandigena* sp. nov. *	GUCC 422.16^T^	–	n/a	n/a	–	–
*Diaporthe juglandigena* sp. nov. *	GUCC 422.161	0	n/a	n/a	0	0
*Diaporthe chimonanthi*	HGUP191001	18	n/a	n/a	n/a	5
*Diaporthe chimonanthi*	HGUP192087	23	n/a	n/a	n/a	2
*Diaporthe caryae*	CFCC 52563^T^	2	n/a	n/a	6	17
*Diaporthe caryae*	PSCG520	4	n/a	n/a	6	30
		**(with gap: 98 characters)**	**(no data)**	**(no data)**	**(with gap: 275 characters)**	**(with gap: 151 characters)**
*Diaporthe eucommiigena* sp. nov. *	GUCC 420.9^T^	–	n/a	n/a	–	–
*Diaporthe eucommiigena* sp. nov. *	GUCC 420.19	0	n/a	n/a	0	0
*Diaporthe passiflorae*	CBS 132527^T^	11	n/a	n/a	23	13
*Diaporthe malorum*	CAA734^T^	15	n/a	n/a	19	13
*Diaporthe malorum*	CAA740	15	n/a	n/a	19	13
*Diaporthe malorum*	CAA752	15	n/a	n/a	19	13
		**(with gap: 128 characters)**	**(no data)**	**(no data)**	**(with gap: 147 characters)**	**(with gap: 255 characters)**
*Diaporthe dejiangensis* sp. nov. *	GUCC 421.2^T^	–	n/a	n/a	–	–
*Diaporthe dejiangensis* sp. nov. *	GUCC 421.21	0	n/a	n/a	0	0
*Diaporthe eres*	CBS 138594^T^	4	n/a	n/a	6	6
*Diaporthe eres*	CAA801	3	n/a	n/a	7	7
*Diaporthe cotoneastri*	CBS 439.82^T^	7	n/a	n/a	5	13
		**(with gap: 67 characters)**	**(no data)**	**(no data)**	**(with gap: 105 characters)**	**(with gap: 138 characters)**
*Diaporthe tongrensis* sp. nov. *	GUCC 421.10^T^	–	n/a	n/a	–	–
*Diaporthe tongrensis* sp. nov. *	GUCC 421.101	0	n/a	n/a	0	0
*Diaporthe phragmitis*	CBS 138897^T^	8	n/a	n/a	n/a	14
		**(with gap: 98 characters)**	**(no data)**	**(no data)**	**(with gap: 145 characters)**	**(with gap: 216 characters)**
*Diaporthe hyperici* sp. nov. *	GUCC 414.4^T^	–	n/a	n/a	–	–
*Diaporthe hyperici* sp. nov. *	GUCC 414.14	0	n/a	n/a	0	2
*Diaporthe caulivora*	CBS 127268^T^	17	n/a	n/a	30	11
*Diaporthe caulivora*	Dip1	17	n/a	n/a	29	n/a
*Diaporthe caulivora*	Dpc11	17	n/a	n/a	29	n/a
		**ITS** **(1–572 characters)**	**LSU** **(573–1423 characters)**	** *rpb2* ** **(1424–2460 characters)**	** *tef1* ** **(2461–2861 characters)**	** *tub2* ** **(2862–3354 characters)**
		**(with gap: 34 characters)**	**(with gap: 11 characters)**	**(with gap: 279 characters)**	**(no data)**	**(no data)**
*Gnomoniopsis rosae*	CBS 145085	–	–	–	(no data)	(no data)
*Gnomoniopsis rosae* *	GUCC 408.7	0	0	0	n/a	n/a
*Gnomoniopsis rosae* *	GUCC 408.17	0	0	0	n/a	n/a
		**ITS** **(1–595 characters)**	**LSU** **(596–1767 characters)**	**(no data)**	** *tef1* ** **(1768–2165 characters**	**(no data)**
		**(with gap: 40 characters)**	**(with gap: 390 characters)**	**n/a**	**(with gap: 135 characters)**	**n/a**
*Coniella quercicola*	CBS 904.69^T^	–	–	n/a	–	n/a
*Coniella quercicola*	CBS 283.76	0	0	n/a	1	n/a
*Coniella quercicola*	CPC 12133	1	n/a	n/a	5	n/a
*Coniella quercicola* *	GUCC 414.2	0	0	n/a	10	n/a
*Coniella quercicola* *	GUCC 414.21	0	0	n/a	10	n/a
*Coniella quercicola* *	GUCC 412.3	0	0	n/a	18	n/a
*Coniella quercicola* *	GUCC 405.6	0	0	n/a	7	n/a
*Coniella quercicola* *	GUCC 405.16	0	0	n/a	6	n/a

## Data Availability

All data generated or analyzed during this study are included in this published article and/or are available from the corresponding author upon reasonable request.

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
