# Peer review of "Taxonomy and Multigene Phylogeny of Diaporthales in Guizhou Province, China"

_jof, 2022, doi:10.3390/jof8121301_

Round 1

Reviewer 1 Report

Review to Manuscript ID: jof-2045053

Title: Taxonomy and phylogeny of Diaporthales in Guizhou Province, China

Authors: Si Yao Wang , Eric HC McKenzie , Alan JL Phillips , Yan Li , Yong Wang

Generally, this paper is of interest. Authors on the basis of multigene phylogenetic analyses present new taxa of Diaporthales: one new genus Pseudomastigosporella gen. nov  and 7 new species belonging to Pseudomastigosporella, Chrysofolia and Daiporte genera. In addition, authors present a new habitat records for Gnomoniopsis rosae and Coniella quercicola . Detailed morphological descriptions and good illustrations of novel taxa including phylogenetic trees of Foliocryphiaceae and Cryphonectriaceae  families and of Daiporte , Gnomoniopsis  and  Coniella genera are provided. I think that results of this study are new and important for taxonomy of Diaporthales, new taxa are good described and analyzed  but not enough discussed. I think that Abstract, Introduction and Discussion sections needs a major revision and correction. Discussion and Conclusions section is very short and must be improved.  I  recommend publishing this paper in ‘Journal of Fungi’ but  after major revision.

Comments to the Author:

I propose to correct the Title: Taxonomy and multigene phylogeny of Diaporthales from Guizhou Province, China

Abstract also must be improved.

Line 16. Please clarify how many species represented 23 isolated strains and also note that this study  was based on multigene phylogenetic analyses.

Line 18. Please clarify which  new genus ‚....a new genus ? in Foliocryphiaceae isolated from Acer palmatum and Hypericum patulum...‘

Line 19. Also you can clarify from which hosts was isolated the new species of Diaporthe.

Keywords: 1new genus Please include the Latin name of novel genus Pseudomastigosporella

Introduction needs some revision and correction.

Please correct this part. It is not necessary in details discuss families and genera of Diaporthales in this part, more detail discussion of taxonomical changes is better transfer to Discussion section. In Introduction this information you can present shortly in several sentences. I propose to insert historical data about studies on Diaporthales from China.

Line 81 ‘……..combined with molecular data and morphology…………’

Clarify all Latin names, must be - in italic font

In the end of introduction please clarify the aim and note how many and which taxa  you present

            Materials and methods

Line 89. The plant samples? It is not clear which part of plants? Dead or live? Infected, symptomatic or with fruitbodies?

Line 90. Please clarify the years when samples were collected.

Line 95. Diaporthe spp.

Line 105-106. ‘…..Morphological observations were made with a Zeiss Scope 5 ? microscope……’

Line 107. ‘…and measurements were made with ZEN 3.0. ? program? It is not clear.

Line 120. Please insert a sentence that sequences of all isolates of all analyzed genes were deposited in GenBank

Results

Line 172. Table 2.  I propose insert also Cryphonectriaceae ‘ Strains and their GenBank accession numbers used in the molecular phylogenetic analyses of Cryphonectriaceae and Foliocryphiaceae…’ because in phyltree I see both families. I think that it will be better to insert into the Table 2, 3, 4  data about yours strains isolated from China and their GenBank accession numbers of sequences (you can separate them in bold font) but not place separately in the Table 6.

Line 175. Please insert  The alignment included 43 taxa, including the represents of Cryphonectriaceae family and outgroup …..’

Line 183.  Please clarify.  ‘..Two new strains of Chrysofolia coriariae sp. nov.collected during this study in Suizhou Province (GUCC 416.4, ex-type strain and GUCC 416.14) shared th

Line 194. Please clarify.  ..’ Four new strains of Pseudomastigosporella guizhouensis sp. nov. from China (GUCC 406.6, ex-type strain, GUCC 405.3, GUCC 405.4 and GUCC 405.8) 19…….’

Line 228.   Two new strains of Diaporthe juglandigena sp.nov. from China (GUCC 422.16, ex-type strain and GUCC 422.161) …’

Line 239. Strains of Diaporthe eucommiigena sp.nov. GUCC 420.9 (ex-type strain) and GUCC 420.19

Line 249. Strains of Diaporthe dejiangensis sp.nov. GUCC 421.2 (ex-type strain) and GUCC 421.21 have a close relationship with

Line 257. Strains of Diaporthe tongrensis sp.nov. GUCC 421.10 (ex-type strain) and GUCC 421.101

Line 265. Two strains of Diaporthe hyperici sp. nov. (GUCC 414.4, ex-type strain and GUCC 414.41)

Line 291. Our strains of Gnomoniopsis rosae (GUCC 408.7 and GUCC 408.17)

Line 314. Strains of Coniella quercicola GUCC 414.2, GUCC 414.21, GUCC 412.3, GUCC 405.6 and GUCC 405.16

Line 343-344.  This sentence you can reject. ‘In this section, we introduce one new genus, seven new species and two new host and country records.

Line 350 Asexual morph: -in bold

The description of new genus and  all new species, I suggest, began with information about the life style of the fungus (saprobic, biotrophic or parasitic) and about which part of the host plants it was found (lives, stalk or fruits and dead or alive).

Line 360. Sexual morph – in bold

Line 368. However, following our Phylogenetic analyses we still propose indicated that Pseudomastigosporella should be placed in Foliocryphiaceae family.

Line 372. Etymology: In reference to the location where the fungus was found, being isolated from Guizhou Province.

Line 417-418. Please reject

Type: China, Guizhou Province, Longli county, on leaves of Coriaria nepalensis, June 2021, S.Y. Wang (HGUP 416, holotype; ex-type living culture GUCC 416.4).

and insert after Description

Line 421. Material examined: China, Guizhou Province, Longli county, on leaves of Coriaria nepalensis, June 2021, S.Y. Wang (HGUP 416, holotype), culture ex-type GUCC 416.4, additional living culture: GUCC 416.14.

The same changes must be done  for all described species.

Discussion and conclusion section must be improved.

I don’t see real discussion. You can discuss phylogenetic relationships of novel taxa and new isolates from China (Gnomoniopsis rosae and Coniella quercicola) and other represents of Foliocryphiaceae, Diaporthaceae, Schizoparmeaceae, Gnomoniaceae with some other related families of Diaporthales, including Cryphonectriaceae.  You can discuss taxonomical changes in Diaporthales and taxonomical position and ecology of new species.  You can discuss hosts, life style and distribution of new species and compare them with other close species.

 Why you note in conclusions eight  new species? In abstract and keywords I see - 7 new species?

 Fig. 10a and Fig.11a - are identical?

Why? Did you isolate different species from the same fruit?

References

Please check all Latin names of species and genera, they must be in italic.

Author Response

Thank you very much for your help to improve our manuscript (jof-2045053). Now we have finished the revision of manuscript and asked for a serious check of English writing by one of the editing services listed at https://www.mdpi.com/authors/english or have your manuscript checked by a native English-speaking colleague. Now I would like to answer the reviewer’s comment one by one:

Reviewer 1

Point 1: I propose to correct the Title: Taxonomy and multigene phylogeny of Diaporthales from Guizhou Province, China

Response 1: Yes, I corrected the title. (As shown in the line 2–3: Taxonomy and multigene phylogeny of Diaporthales in Guizhou Province, China)

Abstract also must be improved.

Point 2: Line 16. Please clarify how many species represented 23 isolated strains and also note that this study  was based on multigene phylogenetic analyses.

Response 2: Yes, I clarified the novel genus name. (As shown in the line 15–16: In a study of fungi isolated from plant material in Guizhou Province, China, we identified 23 strains of Diaporthales belonging to 9 species.)

Point 3: Line 18. Please clarify which new genus ‚....a new genus ? in Foliocryphiaceae isolated from Acer palmatum and Hypericum patulum...‘

Response 3: Yes, I clarified the novel genus name.  (As shown in the line 16–17: The fungi include a new genus (Pseudomastigosporella) in Foliocryphiaceae isolated from Acer palmatum and Hypericum patulum, )

Point 4: Line 19. Also you can clarify from which hosts was isolated the new species of Diaporthe.

Response 4: Yes, I added the hosts for the new species of Diaporthe. (As shown in the line 18–19: and five new species of Diaporthe isolated from Juglans regia, Eucommia ulmoides, and Hypericum patulum. )

Point 5: Keywords: 1 new genus Please include the Latin name of novel genus Pseudomastigosporella

Response 5: Yes, I added the Latin name of novel genus Pseudomastigosporella. (As shown in the line 21–22: Keywords: one new genus; Pseudomastigosporella; seven new species; Chrysofolia; Diaporthe; Foliocryphiaceae)

Point 6: Introduction needs some revision and correction.

Please correct this part. It is not necessary in details discuss families and genera of Diaporthales in this part, more detail discussion of taxonomical changes is better transfer to Discussion section. In Introduction this information you can present shortly in several sentences. I propose to insert historical data about studies on Diaporthales from China.

Response 6: Yes, I modified introduction. (As shown in the line 25–63)

Point 7: Line 81 ‘……..combined with molecular data and morphology…………’

Response 7: I eventually removed this sentence based on the adjustments of Introduction.

Point 8: Clarify all Latin names, must be - in italic font

Response 8: Yes, I corrected them.

Point 9: In the end of introduction please clarify the aim and note how many and which taxa  you present

Response 9: Yes, I added it. (As shown in the line 60–63: The present study follows a recently revised classification [1] combined with molecular data, morphology, and pairwise homoplasy index (PHI) test results and introduces seven novel taxa and two newly recorded taxa within the family Diaporthales found in Guizhou, China.)

Materials and methods

Point 10: Line 89. The plant samples? It is not clear which part of plants? Dead or live? Infected, symptomatic or with fruit bodies?

Response 10: Sometimes leaves or fruits of plants; live; symptomatic. I modified it. (As shown in the line 66–67: The live plant samples were collected from Wengan, Longli, and Dejiang counties, Guizhou Province, China, in June and September 2021 and March 2022.)

Point 11: Line 90. Please clarify the years when samples were collected.

Response 11: Yes, I added it. (As shown in the line 66–67)

Point 12: Line 95. Diaporthe spp.

Response 12: Yes, I added it.

Point 13: Line 105-106. ‘…..Morphological observations were made with a Zeiss Scope 5 ? microscope……’

Response 13: Yes.

Point 14: Line 107. ‘…and measurements were made with ZEN 3.0. ? program? It is not clear.

Response 14: Yes, I added it. (As shown in the line 83–86: Morphological observations were made with a Zeiss Scope 5 (Axioscope 5, China) equipped with an AxioCam 208 color camera (ZEN 3.0) and measurements were made with program (ZEN 3.0). Adobe Photoshop CC 2017 was used to make the photoplates.)

Point 15: Line 120. Please insert a sentence that sequences of all isolates of all analyzed genes were deposited in GenBank

Response 15: Yes, I added it.(As shown in the line 98–99)

Results

Point 16: Line 172. Table 2.  I propose insert also Cryphonectriaceae ‘ Strains and their GenBank accession numbers used in the molecular phylogenetic analyses of Cryphonectriaceae and Foliocryphiaceae…’ because in phyltree I see both families. I think that it will be better to insert into the Table 2, 3, 4  data about yours strains isolated from China and their GenBank accession numbers of sequences (you can separate them in bold font) but not place separately in the Table 6.

Response 16: Yes, I added Cryphonectriaceae. (As shown in the line 158)

Point 17: Line 175. Please insert  ‚The alignment included 43 taxa, including the represents of Cryphonectriaceae family and outgroup …..’

Response 17: Yes, I added Cryphonectriaceae. (As shown in the line 159)

Point 18: Line 183.  Please clarify.  ‘..Two new strains of Chrysofolia coriariae sp. nov. collected during this study in Guizhou Province (GUCC 416.4, ex-type strain and GUCC 416.14) shared th

Response 18: Yes, I clarified it. (As shown in the line 168–169)

Point 19: Line 194. Please clarify.  ..’ Four new strains of Pseudomastigosporella guizhouensis sp. nov. from China (GUCC 406.6, ex-type strain, GUCC 405.3, GUCC 405.4 and GUCC 405.8) 19…….’

Response 19: Yes, I clarified it. (As shown in the line 180–181)

Point 20: Line 228.   Two new strains of Diaporthe juglandigena sp.nov. from China (GUCC 422.16, ex-type strain and GUCC 422.161) …’

Response 20: Yes, I clarified it. (As shown in the line 217–218)

Point 21: Line 239. Strains of Diaporthe eucommiigena sp.nov. GUCC 420.9 (ex-type strain) and GUCC 420.19

Response 21: Yes, I clarified it.  (As shown in the line 66–67)

Point 22: Line 249. Strains of Diaporthe dejiangensis sp.nov. GUCC 421.2 (ex-type strain) and GUCC 421.21 have a close relationship with

Response 22: Yes, I clarified it.  (As shown in the line 240–241)

Point 23: Line 257. Strains of Diaporthe tongrensis sp.nov. GUCC 421.10 (ex-type strain) and GUCC 421.101

Response 23: Yes, I clarified it.  (As shown in the line 248–249)

Point 24: Line 265. Two strains of Diaporthe hyperici sp. nov. (GUCC 414.4, ex-type strain and GUCC 414.41)

Response 24: Yes, I clarified it.  (As shown in the line 257–258)

Point 25: Line 291. Our strains of Gnomoniopsis rosae (GUCC 408.7 and GUCC 408.17)

Response 25: Yes, I clarified it.  (As shown in the line 285–286)

Point 26: Line 314. Strains of Coniella quercicola GUCC 414.2, GUCC 414.21, GUCC 412.3, GUCC 405.6 and GUCC 405.16

Response 26: Yes, I clarified it.  (As shown in the line 311–312)

Point 27: Line 343-344.  This sentence you can reject. ‘In this section, we introduce one new genus, seven new species and two new host and country records.

Response 27: Yes, I deleted it.

Point 28: Line 350 Asexual morph: -in bold

Response 28: Yes, I bolded all in this paper.

Point 29: The description of new genus and  all new species, I suggest, began with information about the life style of the fungus (saprobic, biotrophic or parasitic) and about which part of the host plants it was found (lives, stalk or fruits and dead or alive).

Response 29: Yes, I added the life style of the fungus in this paper.

Point 30: Line 360. Sexual morph – in bold

Response 30: Yes, I bolded all in this paper.

Point 31: Line 368. However, following our Phylogenetic analyses we still propose indicated that Pseudomastigosporella should be placed in Foliocryphiaceae family.

Response 31: Yes, I modified this paragragh. (As shown in the line 356–364:Notes: In Foliocryphiaceae, the important morphological characters of asexual morph was to produce dimorphic conidia. The microconidia were minute, cylindrical, aseptate, hyaline to pale brown; macroconidia were fusoid, aseptate, hyaline [1]. However, Pseudomastigosporella only had macroconidia but like species in Mastigosporellaceae with an apical appendage developing as continuation of conidium body. This feature contradicted the root of“key to genera in Cryphonectriaceae, Foliocryphiaceae, and Mastigosporellaceae”provided by Jiang et al. [1]. However, following our phylogenetic analyses we still proposed that Pseudomastigosporella should be placed in Foliocryphiaceae family.)

Point 32: Line 372. Etymology: In reference to the location where the fungus was found, being isolated from Guizhou Province.

Response 32:Yes, I modified it.(As shown in the line 367–368)

Point 33: Line 417-418. Please reject

Response 33:Yes, I deleted it.

Point 34: Type: China, Guizhou Province, Longli county, on leaves of Coriaria nepalensis, June 2021, S.Y. Wang (HGUP 416, holotype; ex-type living culture GUCC 416.4).

and insert after Description

Line 421. Material examined: China, Guizhou Province, Longli county, on leaves of Coriaria nepalensis, June 2021, S.Y. Wang (HGUP 416, holotype), culture ex-type GUCC 416.4, additional living culture: GUCC 416.14.

Response 34:Yes, I checked and modified all.

Point 35: The same changes must be done  for all described species.

Response 35:Yes, I bolded all in this paper.

Point 36: Discussion and conclusion section must be improved.

I don’t see real discussion.

You can discuss phylogenetic relationships of novel taxa and new isolates from China (Gnomoniopsis rosae and Coniella quercicola) and other represents of Foliocryphiaceae, Diaporthaceae, Schizoparmeaceae, Gnomoniaceae with some other related families of Diaporthales, including Cryphonectriaceae.  You can discuss taxonomical changes in Diaporthales and taxonomical position and ecology of new species.  You can discuss hosts, life style and distribution of new species and compare them with other close species.

Response 36:Yes, I modified it. (As shown in the line 762–826)

Point 37: Why you note in conclusions eight  new species? In abstract and keywords I see - 7 new species?

Response 37:Yes, only 7 new species and I modified it.

Point 38: Fig. 10a and Fig.11a - are identical?

Response 38:Yes, it is same.

Point 39: Why? Did you isolate different species from the same fruit?

Response39:Yes.

Point 40: References

Please check all Latin names of species and genera, they must be in italic.

Response 40:Yes, I checked and modified all.

I look forward to receiving good news from you.

With best regards

Si-Yao Wang

Yong Wang

Reviewer 2 Report

The manuscript is well presented. I do question the use of some of the older background literature such as Alexopoulos and Mims versus more recent texts. Not also that Alexopoulos is misspelled in the references. Also why the Hawksworth edition of Dictionary of Fungi and not the more recent.

The language in the descriptive parts is sometimes difficult or ambiguous. I have appended a version with some comments that might be used as a guide for revising the manuscript.

The different notations of scales on the plates should also be addressed.

Author Response

Thank you very much for your help to improve our manuscript (jof-2045053). Now we have finished the revision of manuscript and asked for a serious check of English writing by one of the editing services listed at https://www.mdpi.com/authors/english or have your manuscript checked by a native English-speaking colleague. Now I would like to answer the reviewer’s comment one by one:

Reviewer 2

Point 1: Line 28.

Response 1:Yes, I modified it to Latin italics.

Point 2: Is this coelomycetous?

Response 2:Yes.

Point 3: Line 63.

Response 3:Yes, I modified it to Latin italics.

Point 4: Line 365: rework the sentence. tenses do not agree.

Response 4:Yes, I modified this paragragh. (As shown in the line 356–364:Notes: In Foliocryphiaceae, the important morphological characters of asexual morph was to produce dimorphic conidia. The microconidia were minute, cylindrical, aseptate, hyaline to pale brown; macroconidia were fusoid, aseptate, hyaline [1]. However, Pseudomastigosporella only had macroconidia but like species in Mastigosporellaceae with an apical appendage developing as continuation of conidium body. This feature contradicted the root of“key to genera in Cryphonectriaceae, Foliocryphiaceae, and Mastigosporellaceae”provided by Jiang et al. [1]. However, following our phylogenetic analyses we still proposed that Pseudomastigosporella should be placed in Foliocryphiaceae family.)

Point 5: Line 367: what it refers to is ambiguous. the conidia or the genus.

Response 5:Yes, I modified this paragragh. (As shown in the line 356–364)

Point 6: Line 363 to 364: not sure what root is. First choice in the key?

Response 6:Yes, I modified this paragragh. (As shown in the line 356–364)

Point 7: Line 382: arising from what base?

Response 7:Arising from the base of wall layers.

Point 8: Line 387: not sure what this looks like.

Response 8:Similar to a spindle shape, but with rounded ends.

Point 9: Line 390 and 558 : under?

Response 9:Yes, I modified it. 

Point 10: Line 402: among

Response 10:Yes, I modified it. 

Point 11: Line 424: globose what?

Response 11:It is Conidiomata.

Point 12: Line 426: again of what, pycnidium?

Response 12:Yes.

Point 13: Line 427: thicker than what?

Response 13:Thicker than it normal thickness.

Point 14: Line 561: not a sentence

Response 14:Yes, I modified it.

Point 15: Line 738: confusing sentence

Response 15:I revised the discussion and conclusion.

Point 16: Line 748: ??? this is an odd phrase

Response 16:I revised the discussion and conclusion.

Point 17: Line 751: these fungi?

Response 17:Yes, these fungi of Diaporthales.

Point 18: Line 755: or some produced spores more easily

Response 18:Yes, but several Diaporthe in this paper are difficult to produce spores.

Point 19: Line 762: delete for

Response 19:Yes, I deleted it.

I look forward to receiving good news from you.

With best regards

Si-Yao Wang

Yong Wang

Round 2

Reviewer 1 Report

Review to Manuscript ID: jof-2045053

Title: Taxonomy and multigene phylogeny of Diaporthales in Guizhou Province, China

Authors: Si Yao Wang , Eric HC McKenzie , Alan JL Phillips , Yan Li , Yong Wang

This paper is of high interest for taxonomy of Diaporthales. Authors on the basis of the multi-locus phylogenetic analysis and morphological examination described one new genus Pseudomastigosporella gen. nov  and 7 new species belonging to Pseudomastigosporella, Chrysofolia and Daiporte genera. In addition, authors present a new habitat record for Gnomoniopsis rosae and Coniella quercicola .Detailed morphological descriptions of novel taxa and good illustrations are provided. I think that results of this study are new, good enough analysed and I recommend publishing this corrected version of the manuscript after minor corrections

Comments to the Author:

Line 320. Please insert Cryphonectriaceae   -‘Table 2. Strains and their GenBank accession numbers used in the molecular phylogenetic analyses of Cryphonectriaceae and Foliocryphiaceae.’

Line 868. ‘Description: Life style: Saprobic, dead woods of Eucommia ulmoides.’

Line 1252-1253. ‘Gnomoniopsis rosae in our study was isolated as asexual morph from Rosa sp. and was newly recorded for China.’

Author Response

Dear Mr. Hank Zhang,

Thank you very much for your help to improve our manuscript (jof-2045053). Now we have finished the revision of manuscript. Now I would like to answer the reviewer’s comment one by one:

Reviewer 2 (round 2)

Point 1: Line 320. Please insert Cryphonectriaceae -‘Table 2. Strains and their GenBank accession numbers used in the molecular phylogenetic analyses of Cryphonectriaceae and Foliocryphiaceae.’

Response 1: Yes, I added it. (As shown in the line 155–156)

Abstract also must be improved.

Point 2: Line 868. ‘Description: Life style: Saprobic, dead woods of Eucommia ulmoides.’

Response 2: Yes, I added it. (As shown in the line 510)

Point 3: Line 1252-1253.‘Gnomoniopsis rosae in our study was isolated as asexual morph from Rosa sp. and was newly recorded for China.’

Response 3: Yes, I added it. (As shown in the line 808–809)

I look forward to receiving good news from you.

With best regards

Si-Yao Wang

Yong Wang
